# Nonlinear Fourier classification of 663 rogue waves measured in the Philippine Sea

**Yu-Chen Lee** [1] *, **Markus Brühl**[2], **Dong-Jiing Doong**[3], **Sander Wahls** [4]

**1** Delft Center for Systems and Control, Delft University of Technology, Delft, CD, The Netherlands,
**2** Ramboll, Hamburg, Germany, **3** Department of Hydraulic and Ocean Engineering, National Cheng Kung University, Tainan, Taiwan, **4** Karlsruhe Institute of Technology, Institute of Industrial Information Technology, Karlsruhe, Germany

* Y.C.Lee-2@tudelft.nl

**Data Availability Statement:** All data files used to generate the results in this paper are available from the 4TU.ResearchData database (DOI: 10.4121/2b8fd219-d16c-4b13-ae01-a70f1e8a01cb).

**Funding:** This work was supported by the European Research Council (ERC) under the

## Abstract

Rogue waves are sudden and extreme occurrences, with heights that exceed twice the significant wave height of their neighboring waves. The formation of rogue waves has been attributed to several possible mechanisms such as linear superposition of random waves, dispersive focusing, and modulational instability. Recently, nonlinear Fourier transforms (NFTs), which generalize the usual Fourier transform, have been leveraged to analyze oceanic rogue waves. Next to the usual linear Fourier modes, NFTs can additionally uncover nonlinear Fourier modes in time series that are usually hidden. However, so far only individual oceanic rogue waves have been analyzed using NFTs in the literature. Moreover, the completely different types of nonlinear Fourier modes have been observed in these studies. Exploiting twelve years of field measurement data from an ocean buoy, we apply the nonlinear Fourier transform (NFT) for the nonlinear Schrödinger equation (NLSE) (referred to NLSE-NFT) to a large dataset of measured rogue waves. While the NLSE-NFT has been used to analyze rogue waves before, this is the first time that it is systematically applied to a large real-world dataset of deep-water rogue waves. We categorize the measured rogue waves into four types based on the characteristics of the largest nonlinear mode: stable, small breather, large breather and (envelope) soliton. We find that all types can occur at a single site, and investigate which conditions are dominated by a single type at the measurement site. The one and two-dimensional Benjamin-Feir indices (BFIs) are employed to examine the four types of nonlinear spectra. Furthermore, we verify on a part of the data set that for the localized types, the largest nonlinear Fourier mode can be attributed directly to the rogue wave, and investigate the relation between the height of the rogue waves and that of the dominant nonlinear Fourier mode. While the dominant nonlinear Fourier mode in general only contributes a small fraction of the rogue wave, we find that soliton modes can contribute up to half of the rogue wave. Since the NLSE does not account for directional spreading, the classification is repeated for the first quartile with the lowest directional spreading for each type. Similar results are obtained.

European Union's Horizon 2020 Research and Innovation Programme (grant agreement No 716669), and also from 2019 Key Fields Scholarship Program, the Ministry of Education, Republic of China (R.O.C.), Taiwan. The funders had no role in study design, data collection and analysis, decision to publish, or preparation of the manuscript.

**Competing interests:** The authors have declared that no competing interests exist.

## Introduction

Rogue waves are rare, suddenly emerging extreme waves with heights of at least twice the significant wave height. They are a danger for ships and offshore structures [1]. There are many different possible mechanisms that can potentially lead to the formation of oceanic rogue waves, such as the random superposition of Stokes waves [2], nonlinear four-wave interactions [3] and modulational instability (MI) [4–6], currents [7], crossing sea states [6, 8] and inhomogeneous sea floors [9–12]. The practical relevance of these individual mechanisms for rogue waves in the actual ocean is still under discussion [13–16].

Rogue waves are usually investigated from a statistical point of view, which makes it possible to estimate how likely rogue waves occur in random seas. Probability distributions for the significant and maximum wave heights have been derived for linear [17–19] and nonlinear cases [20, 21], assuming uni-directionality and narrow-bandedness. The effect of directional spreading and/or larger bandwidths has been investigated as well [22, 23]. There are also empirical distributions that are directly fit to data [24]. See the following papers for a discussion of further literature [25–29].

The statistical approach to rogue waves is widely applied for tasks like the design of ships and offshore structures [30, 31], but it is less useful for forecasting if a rogue wave is actually about to occur soon. This question is essential for the safe operation of ships and offshore structures after they have been deployed [1]. Many different indicators for the forthcoming occurrence of rogue waves, with varying predictive value and computational cost, have been proposed in the literature. Simple standard wave parameters such as the significant wave height or peak period only have value if they are employed in a region-specific manner [32]. The Benjamin-Feir index (BFI), which is obtained by dividing steepness by bandwidth and is related to the onset of the modulational instability, has been found to correlate with enhanced rogue wave occurrence [3, 4, 33], at least for long-crested waves [34]. The BFI has been integrated into the ECMWF freak wave warning system [35], but was found to perform badly on observational data in other works, so that crest-trough correlation has been proposed an alternative [2, 16]. Another indicator for specific analytic unidirectional rogue waves known as Peregrine breathers comes in the form a triangular spectrum that is present during all times of their evolution [36]. Next to simulations [37, 38], the triangular spectrum could also be observed in wave tank experiments [39]. As an alternative to the usual Fourier transform, wavelets have been considered as well [40, 41]. Yet another approach is to propagate measured time series numerically using either physical models or data-driven approaches to predict upcoming rogue waves [42, 43]. Finally, nonlinear Fourier transforms (NFTs; a.k.a. scattering transforms) have been used to investigate and predict rogue waves as well [44–46].

NFTs were invented in the context of the inverse scattering method for solving integrable partial differential equations [47], but they can also be used as signal processing tools [48, 49]. Their unique capabilities include the detection of possibly hidden solitons [50–53], breather components [54, 55] and nonlinear instabilities [44, 56]. This makes them very interesting candidates for the analysis of rogue waves. Different types of NFTs exist for different types of nonlinear dynamics. Recently, the NFT for the Korteweg-de Vries (KdV) equation with vanishing boundary conditions has been applied to a large data set of measurements from a shallow water site in the southern North Sea [57], at which rogue waves occurred more often than expected [58]. Strongly outstanding solitons in the nonlinear spectrum were found to indicate a rogue wave with high probability (Note that the detected solitons were too small to explain the rogue waves on their own. The solitons instead made other, already large waves even larger, turning them into rogue waves). In this paper, we use the NFT for the periodic

nonlinear Schrödinger equation (NLSE) to analyze ocean wave data from a deep water site. Consequently, we denote this method the (periodic) NLSE-NFT.

The NLSE is a classical model for the propagation of weakly nonlinear narrow-banded wave envelopes in deep water in one dimension. The MI of the NLSE manifests itself in that even very small perturbations of simple initial conditions such as sufficiently large plane waves initially grow exponentially [59]. After the initial stage of exponential growth, the nonlinear dynamics damp the growth and lead to the generation of localized structures which can be described by breather solutions of the NLSE [60, 61]. As the name indicates, a breather is a wave packet which "breathes" up and down in space-time domain and propagates with a particular group velocity within a cycle. Some breathe only once. Breathers describe the growth-decay cycle of the MI and are sometimes considered prototypical rogue waves [62–65].

The periodic NLSE-NFT was first developed to construct explicit formulas for periodic solutions of NLSE [66, 67]. To compute it, the linear spectrum of the so-called Zakharov-Sha-bat operator has to be determined. The periodic NLSE-NFT is then obtained by considering different types of eigenvalues (e.g., with periodic or anti-periodic eigenfunctions). For data that is actually governed by the NLSE, the NLSE-NFT has the interesting property that the evolution of the nonlinear spectrum can be performed analytically. The nonlinear Fourier modes provide information about the spectral signatures in nonlinear spectrum and behavior of the nonlinear system. The most important part of the nonlinear spectrum (i.e., the main spectrum that consists of periodic and anti-periodic eigenvalues) is even a constant of motion, i.e., it does not change at all during propagation with respect to the NLSE. In practice, the nonlinear spectrum will change, at least to some degree, as real ocean waves are often broad-banded and directional. This violates the narrow-band spectrum and uni-directionality assumptions behind the NLSE. However, these assumptions are not uncommon. Many classic wave height distributions are e.g. derived under the same assumptions, yet they are widely applied to real-world data [25]. Similarly, the conventional linear Fourier transform, which only solves linear systems exactly, is widely used to analyze nonlinear processes. We therefore argue that the NLSE-NFT can also be used to analyze processes that are not governed by the NLSE, if the results are interpreted with care. Indeed, the NLSE-NFT has successfully been used in this way both in optics [68–71] and in ocean engineering [54, 72, 73]. The evolution of the nonlinear spectrum is no longer trivial in such cases, but one may expect that it is still a better representation than the conventional linear Fourier spectrum, because it accounts for certain nonlinear effects. It is worth mentioning that recent findings from a physical experiment demonstrate that breathers and solitons based on NLSE can propagate under short-crested and directional conditions [74]. A similar nonlinear directional wave group has been found based on a numerical wave model constrained by stereo images [75]. The propagation of rogue waves within envelope solitons through broad-banded water waves has been observed [72]. One experimental result also shows that the breather solution of NLSE can survive in an oppositely propagating regular wave train [76]. These further increase the feasibility of applying this method, and may not be limited by the aforementioned conditions.

In the literature, the periodic NLSE-NFT has been applied to simulated and wave tank data in many studies, e.g. [44, 45, 77]. However, only a very few studies have applied the periodic NLSE-NFT to experimental and measured oceanic rogue waves so far. In Ref. [54], a storm event in the Currituck Sound that included rogue waves was analyzed and found to be dominated by breather components in the nonlinear spectrum. On the other hand, a giant rogue wave measured in the Bay of Biscay during a storm was dominated by stable "Stokes" modes [73], which will be introduced in Sec. Interpretation of the nonlinear Fourier spectrum. In light of these contrasting results, the question arises if rogue waves in the ocean have typical nonlinear spectral signatures. (We also mention that using the vanishing instead of the

periodic NLSE-NFT, several measured rogue waves were found on top of large soliton groups [72, 78]. Breathers can be interpreted as solitons that interact with a specific background, but solitons that interact with a background do not have to be breathers. Therefore, it is not clear what the periodic NLSE-NFT of these rogue waves would look like).

In this paper, we therefore apply the periodic NLSE-NFT to 663 10-minutes samples containing rogue waves from in-situ measurements in the Philippine Sea, which is a section of the western North Pacific Ocean. Our goal is to investigate if the rogue waves at this site exhibit typical nonlinear spectral signatures. We first categorize the nonlinear spectra based on the characteristics of the largest nonlinear Fourier mode. We then investigate if particular characteristics are more common for rogue waves than others, and compare the occurrence of different types of nonlinear spectra for rogue-wave and non-rogue samples. We investigate the relationship between rogue waves in time series and the largest nonlinear Fourier mode using a new approach. Finally, the results are discussed.

## Materials and methods

### Operation of periodic NLSE-NFT

The most popular boundary conditions for the NLSE that allow a solution using the NFT are vanishing and periodic boundary conditions. There are also NFTs for other evolution equations such as the Korteweg–de Vries equation (KdVE). Many studies have been published on the applications of vanishing-boundary NFT for the NLSE in different nonlinear systems such as optical fibres [69, 79], laser radiation [68], and simulated NLSE system [61]. The periodic NFT is more complicated than its vanishing counterpart [80]. More recent studies have extensively applied periodic NFTs for optical systems [49], shallow water waves [51, 53], deep water waves [48, 81] and simulated NLSE system [82].

In hydrodynamics, the focusing NLSE describes the propagation of the complex envelope of unidirectional progressive free-surface waves in deep water ($k_0 h \geq 1.363$), where $k_0$ is the wave number and $h$ is the water depth. We consider the normalized temporal NLSE with periodic boundary conditions of a segment $[0, l]$,

$$iu_X + u_{TT} + 2|u|^2 u = 0, \quad u(X, T + l) = u(X, T), \tag{1}$$

where $u(X, T)$ represents the complex envelope of the wave field, $X$ and $T$ are the normalized forms for space and time, and $l$ is the period of the time series. The equation does not cover loss of energy or wave dissipation. The details of the normalization process will be discussed later.

The NLSE is called *(Lax-)integrable* because it can be represented by two linear eigenvalue equations. The corresponding linear operators are said to form a *Lax pair* [83]. The integrable nature of the NLSE enables one to construct a number of exact solutions known as finite-gap solutions [66]. The complex potential can be reconstructed by the main spectrum $E_k$ and the auxiliary spectrum $\mu_k(0, 0)$ as follows,

$$[\log u(X, T)]_T = 2i \sum_{k=1}^{g} \mu_k(X, T) + 2iK, \quad K = -\frac{1}{2} \sum_{k=1}^{2g+2} E_k. \tag{2}$$

Here the $\mu_k$ are auxiliary functions known as hyperelliptic modes and $E_k$ are certain complex constants that are independent of time and space. The number $g$ of hyperelliptic modes, which is also known as the number of "gaps" or the "genus" in the literature, is assumed to be finite for mathematical reasons. The dynamics of the complex envelope $u(X, T)$ are originally governed by the auxiliary functions $\mu_k$, which evolve on the Riemann surface. A Riemann surface

is a connected one-dimensional complex analytic manifold, which is vital to the studies on the behavior of complex-valued functions [84]. The auxiliary functions satisfy the following equations on the hyperelliptic Riemann surface of the function $\sqrt{P(z)}$:

$$[\mu_k]_T = -2\mathrm{i}\frac{\vartheta_k\sqrt{P(\mu_k)}}{\prod_{j\neq k}(\mu_k - \mu_j)}, \quad P(z) := \prod_{j=1}^{2g+2}\left(z - E_j\right) \tag{3}$$

$$[\mu_k]_X = 4\mathrm{i}\left(\sum_{j=1}^{g}\mu_j + K - \mu_k\right)\frac{\vartheta_k\sqrt{P(\mu_k)}}{\prod_{j\neq k}(\mu_k - \mu_j)} \tag{4}$$

This Riemann surface is a double covering of the complex plane with the pairs of branch points $E_k$. In order to distinguish two different points, the Riemann sheet indices $\vartheta_k = \vartheta_k(X, T) \in \{\pm 1\}$ are used to change the signs corresponding to $P(\mu_k)$ crossing the chosen branch cut of the square root function. Given the constants $E_k$, initial values for auxiliary functions $\mu_k(X_0, T_0)$, the Riemann sheet indices $\vartheta_k = \vartheta_k(X_0, T_0)$ and $u(X_0, T_0)$, we can recover $u(X, T)$ for any desired value of $X$ by solving these specific equations. These values can be obtained from a single time signal $u(X_0, T)$ through spectral analysis of the Zakharov-Shabat operator. See, e.g., [66]. For our study, we used the software library FNFT [85] to compute them. The results presented in this paper were obtained using version 0.4.1 of the library [86]. More details on FNFT are available online at https://github.com/FastNFT/FNFT.

Alternately, it is possible to solve Eqs (3) and (4) analytically using Riemann theta functions in order to implement the inverse periodic NFT. The inverse periodic NFT is a mapping from the main spectrum $E_k$ and initial conditions for the auxiliary spectrum $\mu_k(X_0, T_0)$ and sheet indices $\vartheta(X_0, T_0)$ to a solution of the complex envelope $u(X, T)$. The exact solutions are of the form

$$u(X, T) = u_0\frac{\theta(\mathbf{k}X - \boldsymbol{\omega}T + \boldsymbol{\phi}^- \mid \boldsymbol{\tau})}{\theta(\mathbf{k}X - \boldsymbol{\omega}T + \boldsymbol{\phi}^+ \mid \boldsymbol{\tau})}e^{i(k'X - \omega'T)} \tag{5}$$

where $u_0$ is a scalar constant, $\boldsymbol{\tau}$ is a so-called Riemann matrix, $\mathbf{k}$ is a vector of wave numbers, $\boldsymbol{\omega}$ is a vector of frequencies, $\boldsymbol{\phi}^-$ and $\boldsymbol{\phi}^+$ are two different phase vectors, and $k'$ and $\omega'$ are the physical Stokes wave corrections to the dispersion relation. The Riemann theta function is defined by

$$\theta(\boldsymbol{z} \mid \boldsymbol{\tau}) = \sum_{\mathbf{n}\in\mathbb{Z}^g}\theta_{\mathbf{n}}e^{i\mathbf{n}\cdot\mathbf{z}} \tag{6}$$

where $\theta_{\mathbf{n}} = \exp[i\pi\,\mathbf{n}\cdot\boldsymbol{\tau}\,\mathbf{n}]$ and $\mathbf{n}$ is an integer vector of length $N$. More details on Riemann theta functions and the solution of periodic NLSE can be found in Ref. [48].

The main spectrum consists of the $E_k$, which are connected by curves known as spines. Just like the main spectrum, the spines remain constant during propagation with respect to the NLSE. In principle, spines are redundant because the signal is already uniquely specified by the main and auxiliary spectrum together with the sheet indices. However, they play an important role when the Riemann theta form of the solution in (6) is computed. Spines provide valuable information about the general characteristics of the solution $u(X, T)$, as they impose topological constraints on the trajectories of the auxiliary spectra $\mu_k(X, T)$ [87, p. 49]. Like the main and auxiliary spectrum, the spines can be found by spectral analysis of the Zakharov-Shabat operator.

## Data acquisition and preprocessing

Surface wave data was derived from a buoy operated by the Coastal Ocean Monitoring Center (COMC), National Cheng Kung University (NCKU), Taiwan [88, 89]. The in-situ measurement device is shown in Fig 1a. It is a discus buoy equipped with an accelerometer-tilt-compass (ATC) sensor that was moored at 21.75˚N, 124.12˚E in a water depth of around 5000 m in the Philippine Sea, in the margin of the western North Pacific Ocean. Compared to the relative position of Taiwan, it is located in the southeast of Taitung county. The buoy is called the Taitung Open Ocean buoy. The buoy is 2.5 meters in diameter and 1310 kg in weight, and its power is supplied by batteries and solar panels. Two anemometers are installed on the top of the buoy about 3 meters above the sea level to measure the wind information including wind speed and wind direction. Navigation assistance is provided by radar reflector. Barometers and temperature sensors measure the atmospheric pressure and air temperature, respectively. Acoustic Doppler current profilers (ADCP) are widely used for measuring ocean currents. The ATC sensor measures buoy accelerations, inclinations, and the azimuth at a frequency of 2 Hz. The measurements of accelerations of buoys are well consistent with the surface wave motions.

As shown in Fig 1b, the buoy motions include six degrees of freedom, which are the translational motions (heave, sway and surge) and rotational motions (roll, pitch, and yaw). Each hour, a time series with a length of 10 min (600 s) and a sampling rate of 2 Hz is recorded. Exemplary time series of the buoy accelerations and corresponding motions of pitch, roll, and azimuth are shown in Fig 2a. The vertical acceleration is used to calculate one-dimensional spectra, where the power spectral density (Fig 2b) is calculated by Fourier transform of the

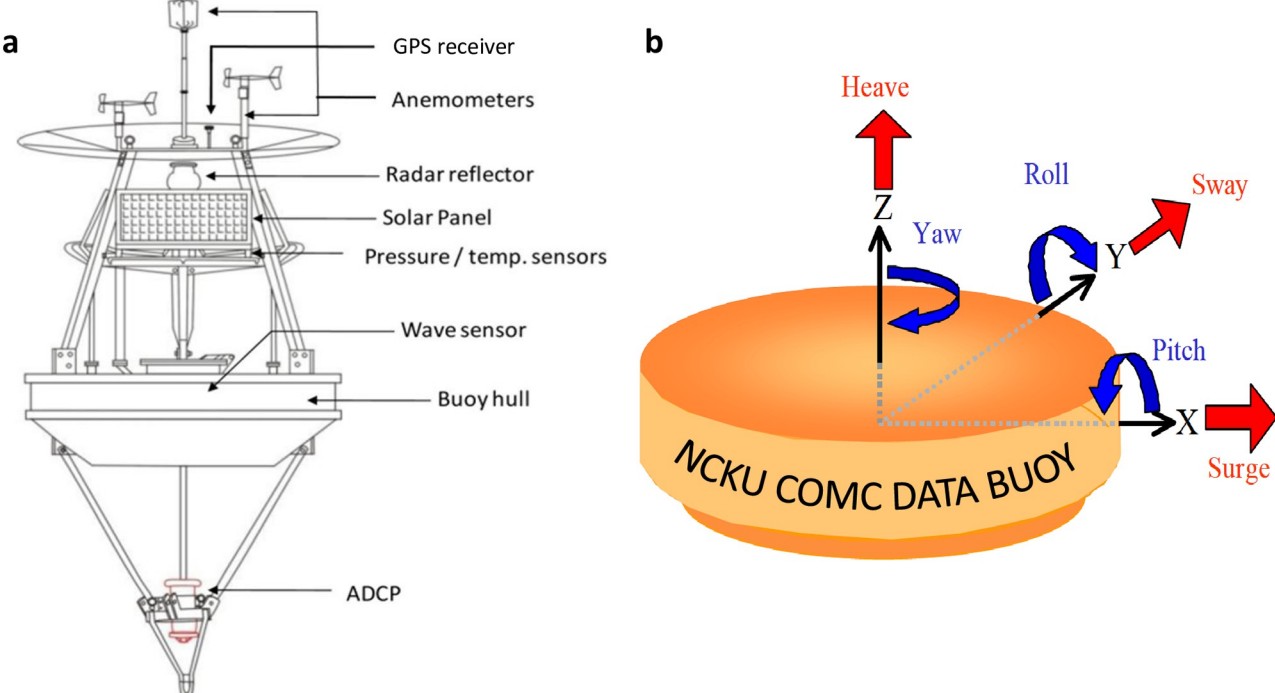

**Fig 1. Measurements of surface waves by wave buoys operated by Coastal Ocean Monitoring Center (COMC), National Cheng Kung University (NCKU), Taiwan.** (a) The discus buoy is equipped with an GPS receiver, anemometers, radar reflector, temperature sensors, accelerometer-tilt-compass (ATC) sensor and acoustic Doppler current profiler (ADCP). (b) The buoy motions by six degrees of freedom are heave, sway and surge, roll, pitch, and yaw.

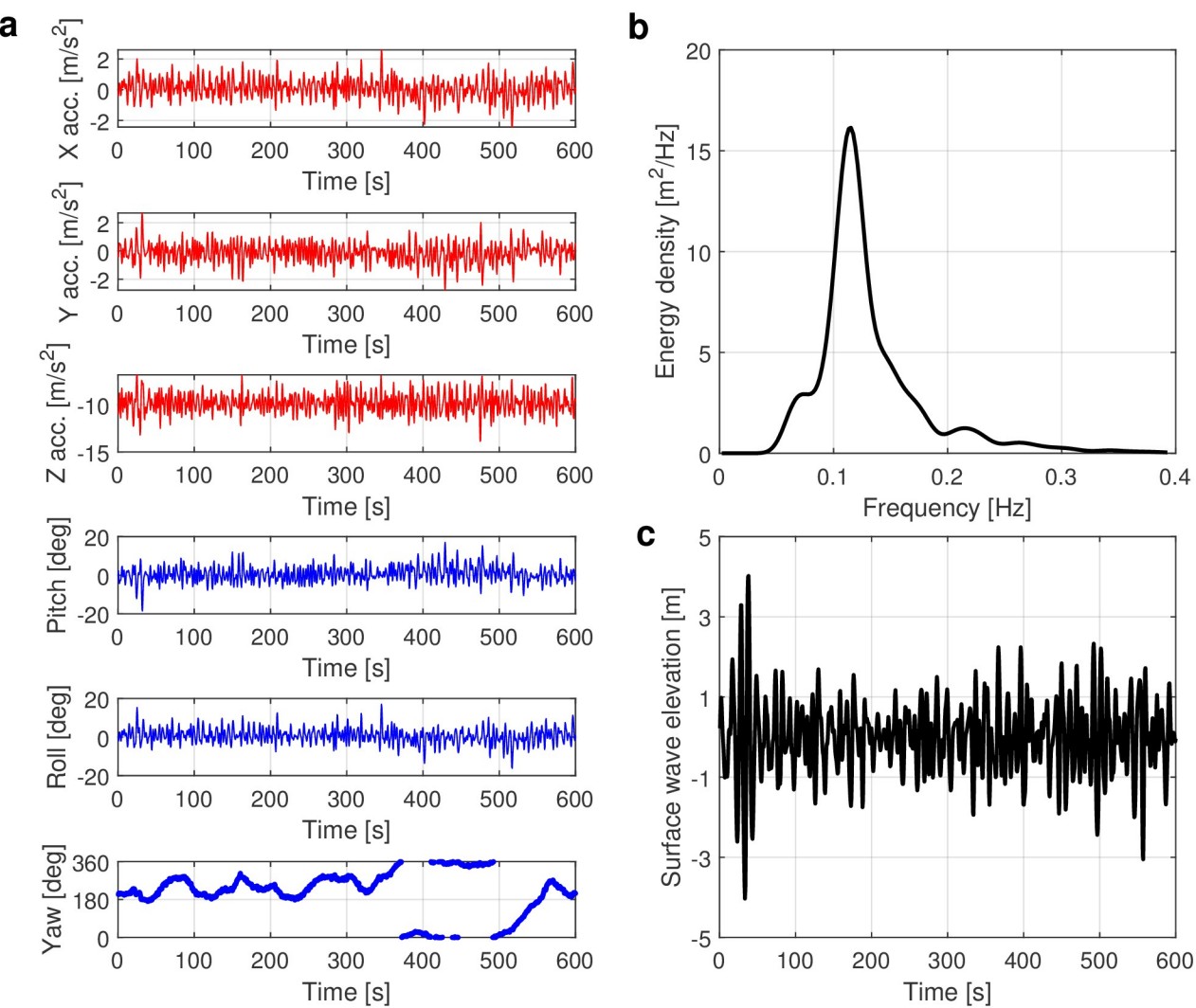

**Fig 2. Signals from measurements.** (a) The time series of buoy accelerations and corresponding motions of pitch, roll, and yaw. (b) The power-density spectrum determined by heave acceleration. (c) The time series of surface-wave elevation determined by wave spectrum.

time series of buoy accelerations. Finally, the surface wave elevation (Fig 2c) is obtained by filtering the low-frequency noise generated during the transformation between acceleration and spectrum [90, 91]. Note that the influence of wave transformation over bathymetry and the effects of refraction can be neglected in our data due to the deep water depth of 5000 m.

To give an idea of what the surface waves in our measurements look like, we provide histograms of basic wave parameters for both rogue and non-rogue waves samples in Fig 3. The wave parameters include significant wave height $H_{1/3}$, maximum wave height $H_{max}$, peak period $T_p$ and directional spreading $\sigma_\theta$. The samples have a wide range of significant wave height $H_{1/3}$ from 0.04 m to 15.28 m, maximum wave height $H_{max}$ from 0.1 m to 28.83 m and peak period $T_p$ from 4.62 s to 21.43 s. This data set covers most of the local environmental conditions and contains most types of waves. The directional spreading is calculated based on the Ref. [92]. Our buoy is located in the open sea, where the waves can come from any direction. The directional spreading of both rogue and non-rogue samples consequently ranges from 52˚ to 83˚ and is therefore quite large.

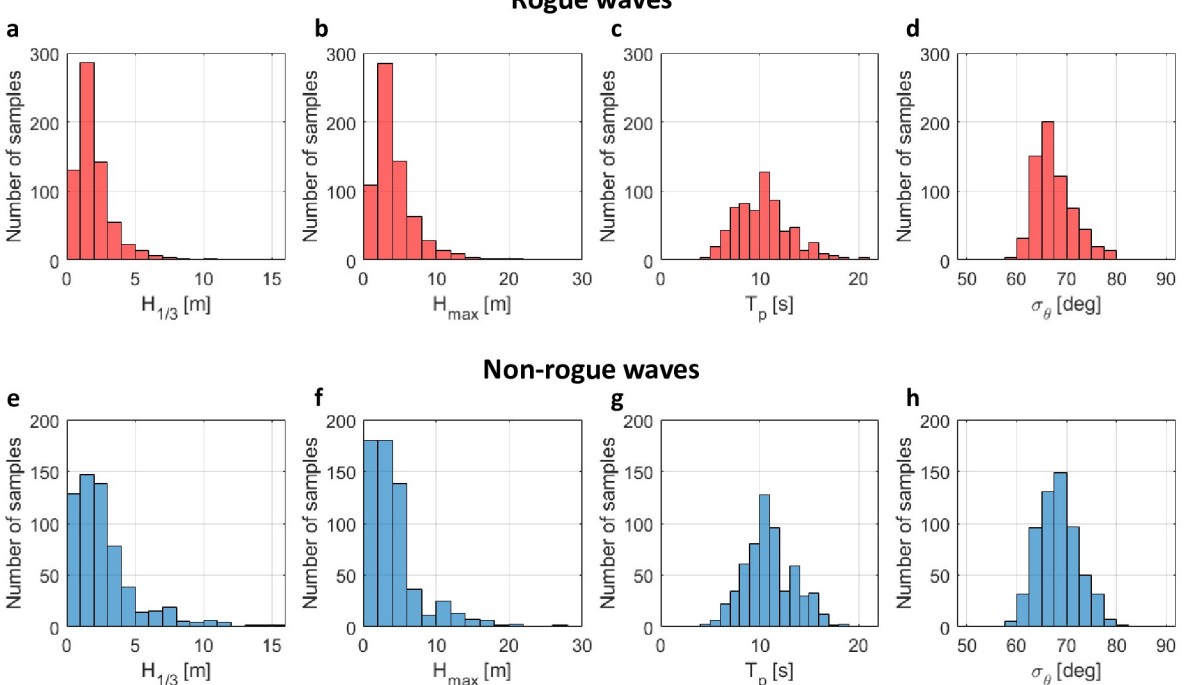

**Fig 3. Histograms of basic wave parameters of measured surface waves.** (a-d) Histograms of rogue wave samples of the significant wave height $H_{1/3}$, maximum wave height $H_{max}$, peak period $T_p$ and directional spreading $\sigma_\theta$, respectively. (e-h) Histograms of non-rogue wave samples of significant wave height $H_{1/3}$, maximum wave height $H_{max}$, peak period $T_p$ and directional spreading $\sigma_\theta$, respectively. There are 663 samples for rogue waves and 600 samples for non-rogue waves, respectively.

## Data normalization of deep water waves for the nonlinear Fourier transform

The field-measurement data has to be processed before it can be analyzed by the NLSE-NFT [81]. The selected samples have to meet the deep-water conditions for validity of the (focusing) NLSE, $k_0 h \geq 1.363$. The NLSE governs the propagation of the complex wave envelope. Consequently, the Hilbert transform is used to calculate the complex wave envelope from the wave-surface elevation after removal of the carrier wave [93]. The Hilbert transform method is accurate to the same order as the NLSE for narrowband wave-trains [94 Eq. 38], and thus commonly used for NLFT-NSE analyses [48, 73, 95]. However, we remark that improved methods for the computation of the envelope, in particular for rogue waves, have been proposed [96, 97].

The unnormalized NLSE describes the evolution of the envelope,

$$i[C_g^{-1}A_t + A_x] + \mu C_g^{-3}A_{tt} + \nu C_g^{-1}|A|^2 A = 0, \tag{7}$$

where A is the complex envelope and $C_g$ is the group velocity of a wave packet. With the dispersion relation $\omega_0^2 = gk_0\sigma$, $\sigma = \tanh k_0 h$, the values of the coefficients for gravity waves in finite water depth $h$ are given by [98]

$$C_g = \frac{c}{2}\left[1 + \frac{(1-\sigma^2)k_0 h}{\sigma}\right], \quad \text{where } c = \frac{\omega_0}{k_0}, \tag{8}$$

$$\mu = \frac{-g}{8k_0 \sigma \omega_0}\{[\sigma - k_0 h(1 - \sigma^2)]^2 + 4k_0^2 h^2 \sigma^2 (1 - \sigma^2)\},\tag{9}$$

$$v = \frac{-k_0^4}{2\omega_0}\left(\frac{c}{2\omega_0}\right)\left\{\frac{(9 - 10\sigma^2 + 9\sigma^4)}{2\sigma^2} + \frac{4c^2 + 4(1 - \sigma^2)cC_g + gh(1 - \sigma^2)^2}{C_g^2 - gh}\right\}.\tag{10}$$

The product of the depth-dependent coefficients should be greater than 0, $\mu v > 0$ (or $k_0 h \geq 1.363$), in order to arrive at (1). To proceed from (7) to (1), we apply the following change of coordinates:

$$X = \frac{\mu}{C_g^3}x, \quad T = t - \frac{x}{C_g}, \quad u(X, T) = \rho A(X, T), \quad \rho = \sqrt{\frac{C_g^2 v}{2\mu}}.\tag{11}$$

The parameter $\rho$ is called the nonlinear parameter by Osborne [48]. By substituting $A = \rho^{-1}u$ in the temporal NLSE, one obtains the normalized temporal NLSE (1). See, e.g., Ref. [81]. Please note that the parameter $\rho$ is not dimensionless. The NLSE (1) is normalized in the sense that its coefficients are independent of physical parameters such as the water depth.

## Rogue waves and sea state parameters

In this study, each 10 min time series has been analysed in time domain by the downward zero-crossing method [99]. This method can extract individual wave heights and wave periods by identification of the water surface crossing the mean water level in an downward direction. Subsequently, each individual wave was ranked and the mean wave height of the one-third largest waves is defined as the significant wave height $H_{1/3}$. The wave are classified as rogue waves if

$$\mathrm{AI} = \frac{H}{H_{1/3}} \geq 2,\tag{12}$$

where the ratio AI is called the abnormality index [100].

To characterize the sea states in which rogue waves occur, we now introduce several parameters. Following [92, 101], the directional spreading is calculated as

$$\sigma_\theta = \left[2\left\{1 - \left(\frac{p^2 + q^2}{m_0^2}\right)^{\frac{1}{2}}\right\}\right]^{\frac{1}{2}},\tag{13}$$

where $p = \int_0^{2\pi}\int_0^\infty \cos(\theta)S(\omega, \theta)d\omega d\theta$, $q = \int_0^{2\pi}\int_0^\infty \sin(\theta)S(\omega, \theta)d\omega d\theta$, and $m_0 = \int_0^{2\pi}\int_0^\infty S(\omega, \theta)d\sigma d\theta$ is the zeroth spectral moment. $S(\omega, \theta)$ is the directional spectrum.

The Benjamin–Feir index (BFI) is a parameter used to describe kurtosis in water waves that can be related to the modulational instability [3]. It is given by the ratio of the spectral steepness $\epsilon$ and the spectral width $\delta_\omega$ for one-dimensional wave spectra,

$$\mathrm{BFI}_{1D} = \frac{\sqrt{2}\epsilon}{\delta_\omega}.\tag{14}$$

The spectral width $\delta_\omega$ can be calculated from Goda's peakedness parameter $Q_p$ [102], which is

related to half-width at half-maximum of the spectrum:

$$\delta_\omega = \frac{1}{Q_p \sqrt{\pi}}, \tag{15}$$

$$Q_p = \frac{2}{m_0^2} \int \omega S^2(\omega) d\omega. \tag{16}$$

The one-dimensional BFI does not account for directionality. Mori et al. (2011) investigated the relationship between wave height distribution and kurtosis in directional seas and introduced a two-dimensional BFI to assess kurtosis behavior within directional sea states [103]. The proposed two-dimensional BFI is given by

$$\mathrm{BFI}_{2D}^2 = \frac{\mathrm{BFI}_{1D}^2}{1 + \alpha_2 R}, \tag{17}$$

where $\alpha_2$ is an empirical constant equal to 7.1, and

$$R = \frac{1}{2} \frac{\delta_\theta^2}{\delta_\omega^2} \tag{18}$$

is the ratio of the directional bandwidth $\delta_\theta$ and the frequency width $\delta_\omega$. Here, the directional bandwidth $\delta_\theta$ is calculated as [35]

$$\delta_\theta = \sqrt{2(1 - M_1)}, \tag{19}$$

$$M_1 = \frac{1}{m_0} \int \int \cos\left(\theta - \theta_p(\omega)\right) S(\omega, \theta) d\omega d\theta, \tag{20}$$

$$\theta_p(\omega) = \arctan\left(\frac{\int \sin(\theta) S(\omega, \theta) d\theta}{\int \cos(\theta) S(\omega, \theta) d\theta}\right) \tag{21}$$

with $\theta_p$ denoting the peak direction.

## Results

### Interpretation of the nonlinear Fourier spectrum

Before we introduce the four classes of nonlinear spectra mentioned in the abstract, we start with a more fundamental study of different characteristic solutions of the NLSE using numerical simulations. The main spectrum and spines in the nonlinear spectrum stay invariant during the evolution with respect to the NLSE. This distinctive feature enables the detection of hidden characteristics such as e.g. envelope solitons (when the NLSE holds). This is the key advantage of the NFT: it describes solutions of NLSE in terms of analytically evolving nonlinear spectral components.

The well-known breather solutions of the NLSE include Akhmediev breathers, Peregrine breathers and Kuznetsov-Ma breathers. These solutions stand as exemplary prototypes for the modeling of rogue waves in various nonlinear media. Recently, large families of breathers that contain these solutions as special cases were investigated in the Refs. [104, 105]. There are two kinds of doubly periodic solutions, A-type and B-type, which differ by the phase shift of their consecutive maxima. The local maxima of A-type breathers occur periodically with a shift of half the temporal period after propagating half the spatial period. B-type breathers exhibit local

maxima that occur periodically without a shift in the temporal period. A- and B-type breathers are specified by parameters known as $\alpha_3$, $\rho$, and $\eta$, and $\alpha_1$, $\alpha_2$, and $\alpha_3$, respectively.

Fig 4 shows four characteristic wave structures in the spatial-temporal domain and in the nonlinear Fourier domain: a constant envelope (first row), a soliton (second row), an A-type breather with parameters $\alpha_3 = 1$, $\rho = 3$, $\eta = 0.4$ (third row) and a B-type breather with parameters $\alpha_1 = 0.1$, $\alpha_2 = 0.5$, $\alpha_3 = 1$ (fourth row). Note that $u(X, T)$ refers to the complex envelope of the surface elevation with $X$ and $T$ denoting normalized space and moving time variables. We compute the nonlinear spectra of these general wave structures using the periodic NLSE-NFT. The corresponding nonlinear spectra of the four characteristic waveforms are shown in the Fig 4b, 4e, 4h and 4k. Note that the nonlinear spectra are symmetric with respect to the real axis. The complex points marked with blue circles in the plots indicate the main spectrum. The red curves connecting pairs of main spectrum points are the spines.

We now discuss the connection between spines and linear stability. It can happen that one spine ends exactly where another begins, so that two points in the main spectrum share the same position. Such points are then called *double points*. For the constant envelope in Fig 4b, a double point occurs at $\lambda = \pm 0.81i$ because two spines touch. For the soliton in Fig 4e, $\lambda = \pm 0.5i$ appears to be a double point, but we know from theoretical results that the spine is just very short. If a double point traps a point in the auxiliary spectrum, it is called *degenerate* [106]. The point $\lambda = 0.86i$ in Fig 4b is an example for a degenerate point. This can be seen in Fig 4c, where the values of the hyperelliptic modes $\mu_k(0, T)$ have been marked with small black dots for $0 \leq T \leq l$. We see that a single black point is stuck at the double main spectrum point at $\lambda = 0.86i$. This shows that the corresponding auxiliary spectrum cannot move away and is thus trapped: $\mu_k(0, 0) = \mu_k(0, T)$ for $0 \leq T \leq l$. Therefore, we conclude that $\lambda = 0.86i$ is degenerate. We can also see that the point $\lambda = 0.5i$ in Fig 4e is not a degenerate double point, because the corresponding hyperelliptic mode in Fig 4f is moving around. It is therefore not degenerate. The A-type breather has double points at the centers of the horizontal spines in Fig 4h. From Fig 4i, we see that they are not degenerate either. The B-type breather also has no degenerate points in Fig 4k. We know this because degenerate points can only occur on spines or on the real axis [56, p. 825].

Degenerate double points in the main spectrum are known to play a fundamental role for the modulational instability [56, 107]. Any double point in the main spectrum will, except in very special cases, split into two single points when the solution is perturbed. If a hyperelliptic mode is trapped by a degenerate double point, it can thus be freed by arbitrarily small perturbations. The freed hyperelliptic mode will sometimes grow rapidly as the waveform evolves and then significantly change the solution via Eq (2). In such cases, the solution turns out to be linearly unstable [56]. The plane wave in Fig 4a is for example known to be linearly unstable [59], which is caused by the degenerate points on the spine [56, 107]. Degenerate points are thus indicators for potential linear instabilities. However, not all degenerate points give rise to instabilities. For example, there are infinitely many degenerate points on the real axis (not shown in Fig 4) that are known not to cause instability [56]. The most useful consequence of this discussion probably is that finite genus solutions of the NLSE without non-real main points in the main spectrum that are located on spines are linearly stable [56].

Following the terminology in [48], spines in the nonlinear spectrum are also called nonlinear Fourier modes. A spine that crosses the real axis (such as the central spines in the Fig 4b, 4h and 4k) is called a stable mode. Stable modes correspond to sine or Stokes waves [48]. (The Stokes waves turn into sine waves for small amplitudes. The spines become vertical lines in that case. See e.g. [54, Fig. 13] or compare Figs 3 and 4 in [86].) A spine that does not cross the real axis (such as the nearly overlapping double points with invisible spines in Fig 4d, the two horizontal spines in Fig 4f and the two vertical spines in Fig 4h that do not cross the real axis)

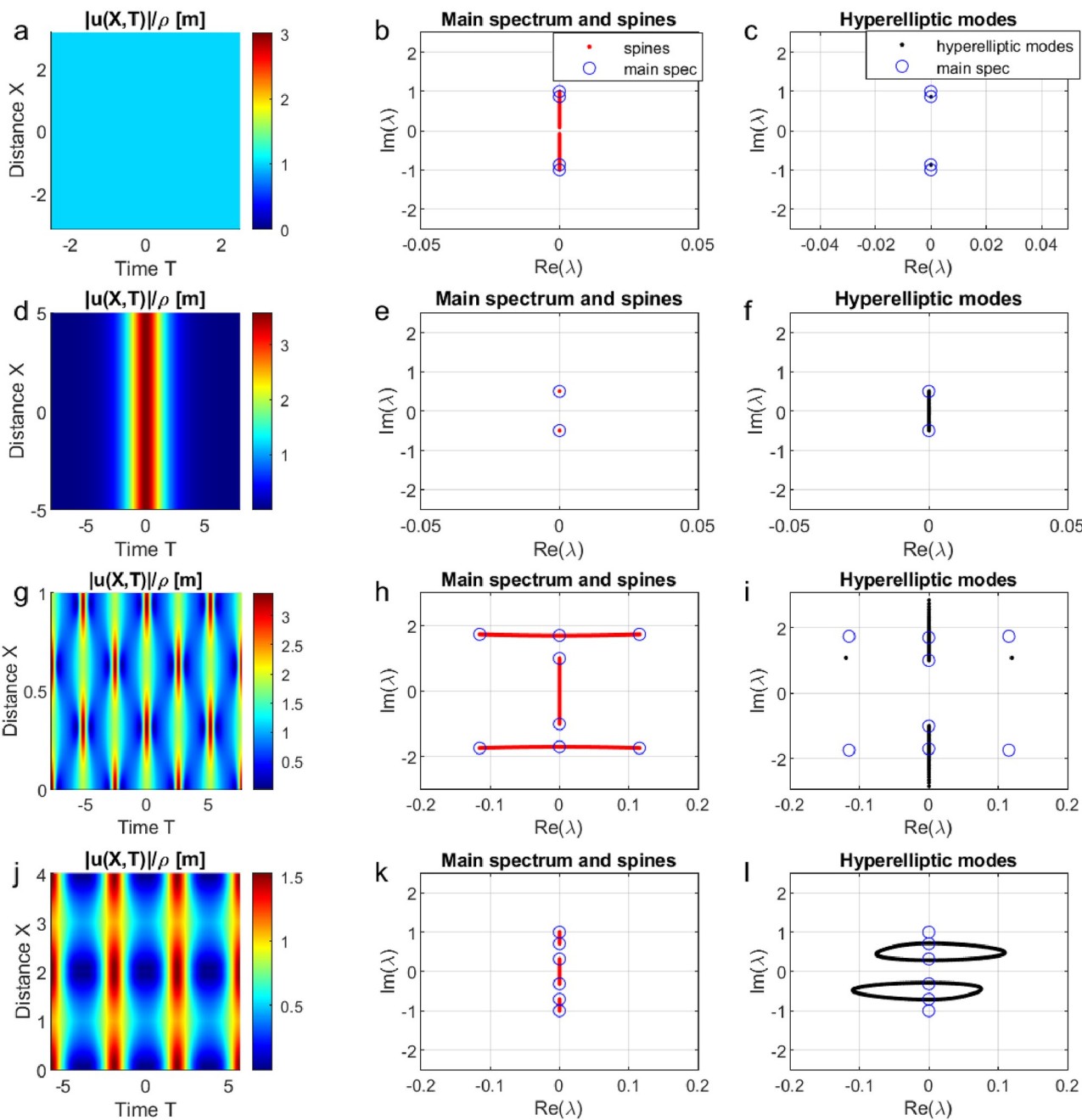

**Fig 4. Characteristic wave structures in the spatial-temporal domain and their analytical results from the nonlinear Fourier transform.** (a) Spatial-temporal evolution of the constant envelope. (b) Main spectrum and spines of the constant envelope. The small gap in the spine at zero is a numerical artifact. (c) Hyperelliptic modes of the constant envelope. (d) Spatial-temporal evolution of the fundamental soliton. (e) Main spectrum and spines of the fundamental soliton. (f) Hyperelliptic modes of the fundamental soliton. (g) Spatial-temporal evolution of the A-type doubly periodic solutions of the NLSE with the parameters $\alpha_3 = 1$, $\rho = 3$, $\eta = 0.4$. (h) Main spectrum and spines of the A-type doubly periodic solutions of the NLSE. (i) Hyperelliptic modes of the A-type doubly periodic solutions. The two individual black points in the upper left and right are numerical artifacts. (j) Evolution of the B-type doubly periodic solutions of the NLSE with the parameters $\alpha_1 = 0.1$, $\alpha_2 = 0.5$, $\alpha_3 = 1$. (k) Main spectrum and spines of the B-type doubly periodic solutions of the NLSE. (l) Hyperelliptic modes of the B-type doubly periodic solutions.

is called an unstable mode [48, 54]. When an unstable mode interacts with a stable mode as in the Fig 4h and 4k, it corresponds to a breather [48]. An unstable mode is considered to be a soliton when, as in Fig 4e, it visually appears to be a double point with a spine of length zero. In that case, no interacting stable mode is needed. Unstable modes are called unstable because they describe the *nonlinear* stage of the modulational instability, which is known to be dominated by breathers [48, p. 276], [106]. Solitons are also counted in this class as they can be obtained by taking the appropriate limits for breather solutions. Technically, the solitons in the periodic NLSE-NFT spectrum are just breathers whose parameters are very close to this limit.

## Four types of nonlinear spectra of rogue wave samples in the ocean

We are now ready to introduce the four different types of nonlinear spectra used in our classification. To illustrate them, we selected four rogue wave records with similar significant wave height $H_{1/3} \cong 3.5$ m. They were measured during storm events with extreme environmental conditions. The four time series with rogue waves have similar significant wave heights $H_{1/3}$ equal to 3.24 m, 3.44 m, 3.62 m and 3.56 m, but different maximum wave heights $H_{max}$ equal to 6.67 m, 6.96 m, 7.53 m, and 8.05 m, respectively. They are shown in Fig 5a to 5d (black lines). The abnormality index (*AI*) is calculated as the ratio of maximum wave height $H_{max}$ to significant wave height $H_{1/3}$ and equals 2.06, 2.02, 2.08 and 2.26, respectively. This qualifies them as rogue waves. We have labelled plus/minus the significant wave height $H_{1/3}$ on the right y-axis, and indeed found the wave elevation range larger than twice of the significant wave height $H_{1/3}$. The red lines in Fig 5 represent the magnitudes of the complex envelopes obtained from the Hilbert transform.

Subsequently, we show the nonlinear spectra of the four rogue wave time series data obtained by using the NLSE-NFT. The nonlinear spectra are presented in Fig 6, where the horizontal axis is the wave frequency in Hz and the vertical axis is the spectral amplitude. Note that the units of the nonlinear spectra in Fig 6 have been chosen such that the NFT reduces to the usual Fourier transform for low amplitude signals. Nonlinear modes however have no interpretation in terms of frequency. In the following, we classify the nonlinear spectra with rogue waves into four types according to their spectral portraits.

All spectral modes of the spectrum in Fig 6a are connected to the real axis. This means we consider them stable modes, which correspond to be either sine or Stokes waves [48]. The difference between the spectral components of sine waves and Stokes waves is that the main spectrum points of sine waves are connected by vertical spines, and the main spectrum points of Stokes waves are connected by distorted spines. In this case, the rogue wave is generated by nonlinear interactions of sine-wave or Stokes-wave components [48]. We categorize this spectrum as type 1, i.e. a stable-mode spectrum.

In Fig 6b, the two points at the coordinates (0.102 Hz, 0.329 m) and (0.103 Hz, 0.383 m) are connected by a spine that does not cross the real axis. Together with the closest stable mode, it represents a breather, which is a combination of a stable and an unstable mode (see Fig 4h and 4k). It indicates some nonlinear effects in the sea state. Here, the unstable mode has an amplitude similar to the stable modes. Since the maximum amplitude of the unstable mode does not exceed 120% of the maximum amplitude of the stable modes, we categorize this spectrum as type 2, i.e. a small-breather spectrum. This threshold is determined through empirical evaluation, which relies on comparing the amplitudes of stable modes and small breathers.

The nonlinear spectrum in Fig 6c has one large breather in the peak center of the spectrum and three small breathers. The largest unstable mode consists of two main spectral points at (0.096 Hz, 0.610 m) and (0.096 Hz, 0.625 m) at the peak of the spectrum. Since the maximum

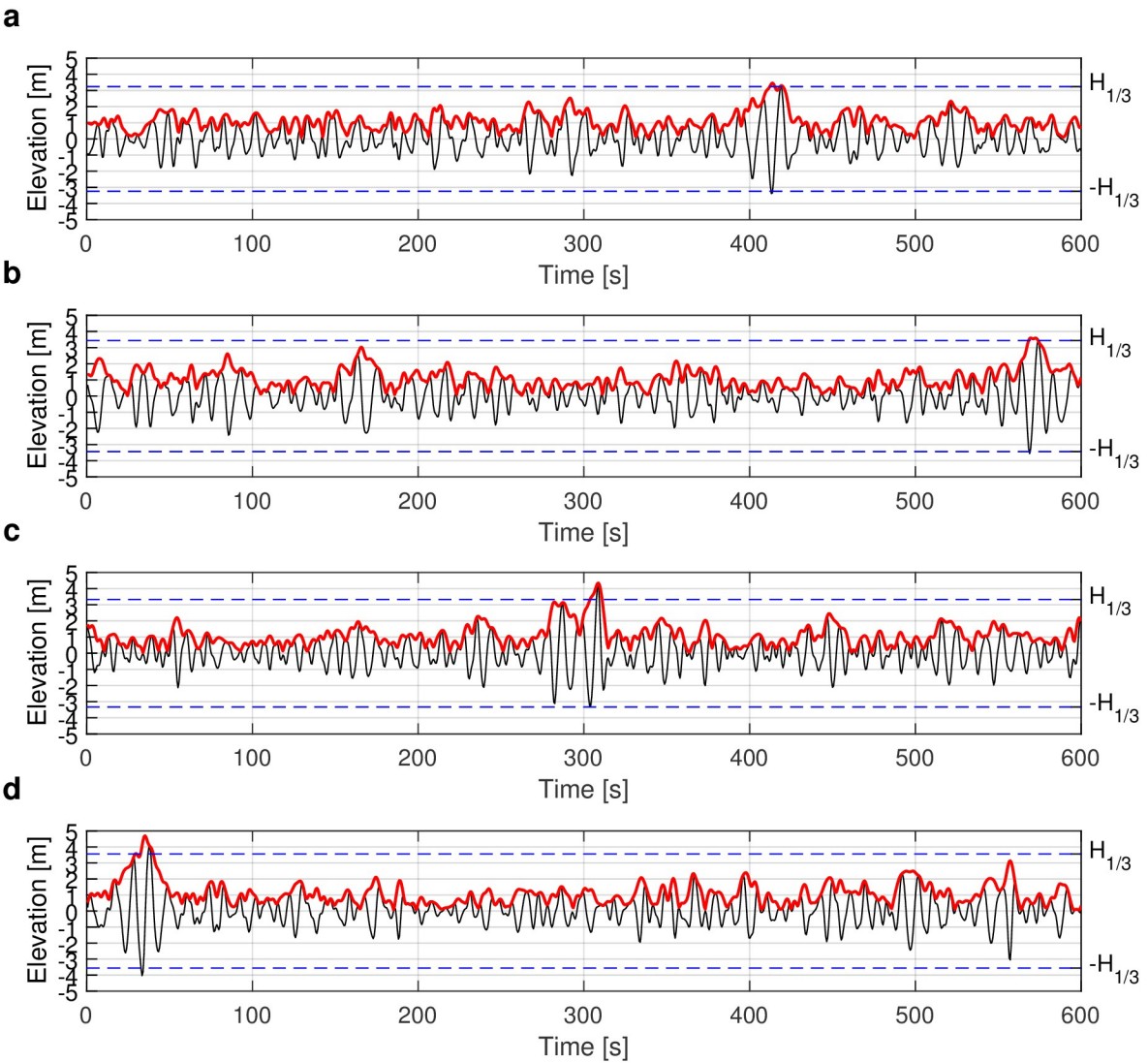

**Fig 5. Time series of rogue wave records from Taitung Open Ocean buoy.** The surface-wave elevation is shown as the black line and the magnitude of the complex envelope calculated by Hilbert transform is shown as red line. The blue dashed lines refer to $\pm H_{1/3}$. (a) The rogue-wave data measured from 20:00h on 18 January 2013 with $H_{max}$ = 6.67 m, $H_{1/3}$ = 3.24 m, and AI = 2.06. (b) The rogue-wave data measured from 22:00h on 4 October 2014 with $H_{max}$ = 6.96 m, $H_{1/3}$ = 3.44 m, and AI = 2.02. (c) The rogue-wave data measured from 08:00h on 4 December 2015 with $H_{max}$ = 7.53 m, $H_{1/3}$ = 3.62 m, and AI = 2.08. (d) The rogue-wave data measured from 20:00h on 25 September 2012 with $H_{max}$ = 8.05 m, $H_{1/3}$ = 3.56 m, and AI = 2.26.

amplitude of the unstable mode is larger than 120% of the maximum amplitude of the stable modes, we categorize this spectrum as type 3, i.e. a large-breather spectrum. The nonlinear spectrum in Fig 6c indicates that the time series of Fig 5c is generated by the nonlinear interactions of one larger breather, three small breathers and other small-amplitude stable modes.

In the nonlinear spectrum in Fig 6d, there is one large breather with two points of main spectrum very close to each other such that the spine reduces to a single red dot in the figure. This constellation corresponds to a periodicized soliton (see Fig 4e). An unstable mode was considered a soliton if, based on visual inspection, the spine reduced to a single point in the nonlinear spectrum. The spectrum thus contains one soliton with two main spectral points around the coordinates (0.091 Hz, 0.778 m), and two small breathers to the right of the soliton.

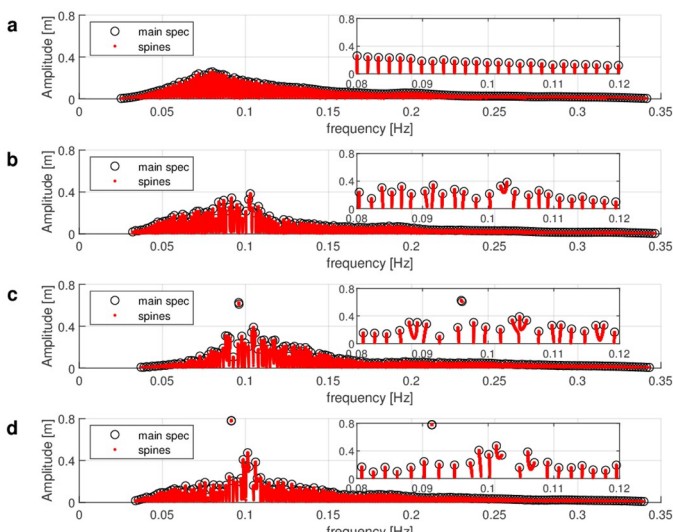

**Fig 6. Nonlinear spectra with envelope amplitudes on the vertical axis and carrier wave frequencies on the horizontal axis.** (a) Spectrum of the time series of complex envelope from Fig 5a is classified as type 1: stable mode. (b) Spectrum of the time series of complex envelope from Fig 5b is classified as type 2: small breather. (c) Spectrum of the time series of complex envelope from Fig 5c is classified as type 3: large breather. (d) Spectrum of the time series of complex envelope from Fig 5d is classified as type 4: soliton.

Since the nonlinear mode with the highest amplitude is a soliton, we categorize the nonlinear spectrum as type 4, i.e. a soliton spectrum.

## Data analysis of wave samples with four types of nonlinear spectra: Stable, small and large breather, soliton

The surface wave data discussed in this paper were measured by the Taitung Open Ocean buoy. There are in total 43018 samples (i.e. time series) in the period from 2006 to 2017. Among these, 663 samples contained rogue waves. In Fig 7a, we plot the peak period $T_p$ obtained via conventional frequency-domain analysis over the maximum wave height $H_{\max}$ for all 663 samples with rogue-wave events. The peak period $T_p$ is defined as the dominant wave period in the wave-energy spectrum. The rogue-wave events in the plot have thereby been classified based on the four types of nonlinear spectra as discussed in the previous subsection. The four types of nonlinear spectra from stable-modes type to soliton type account for 75.1%, 6.3%, 12.2% and 6.3% of the rogue-wave samples, respectively.

Our results show that most of the rogue-wave samples are of the stable-mode type. Under the assumption that the NLSE describes the evolution of the time series at least approximately, this implies that the nonlinear superposition of sine waves or Stokes waves is the main generation mechanism in the real ocean, in accordance with the findings from Ref. [108]. In any case, it does not suggest that breather solutions of the NLSE are prototypical rogue waves in our samples. However, 24.9% of the nonlinear spectra of rogue waves contain unstable modes such as breathers or solitons. This is especially for rogue-wave samples with maximum wave heights larger than 15 m. The results show that the nonlinear spectra of samples with larger maximum wave height almost certainly contain unstable modes. A large portion (90.9%) of the samples with maximum wave height higher than 15 meters is strongly linked to type 3 (large breather spectra) and type 4 (soliton spectra).

Another feature of the four types of rogue-wave spectra can be observed from their peak periods. In general, the larger amplitudes of extreme waves require a large enough wave period

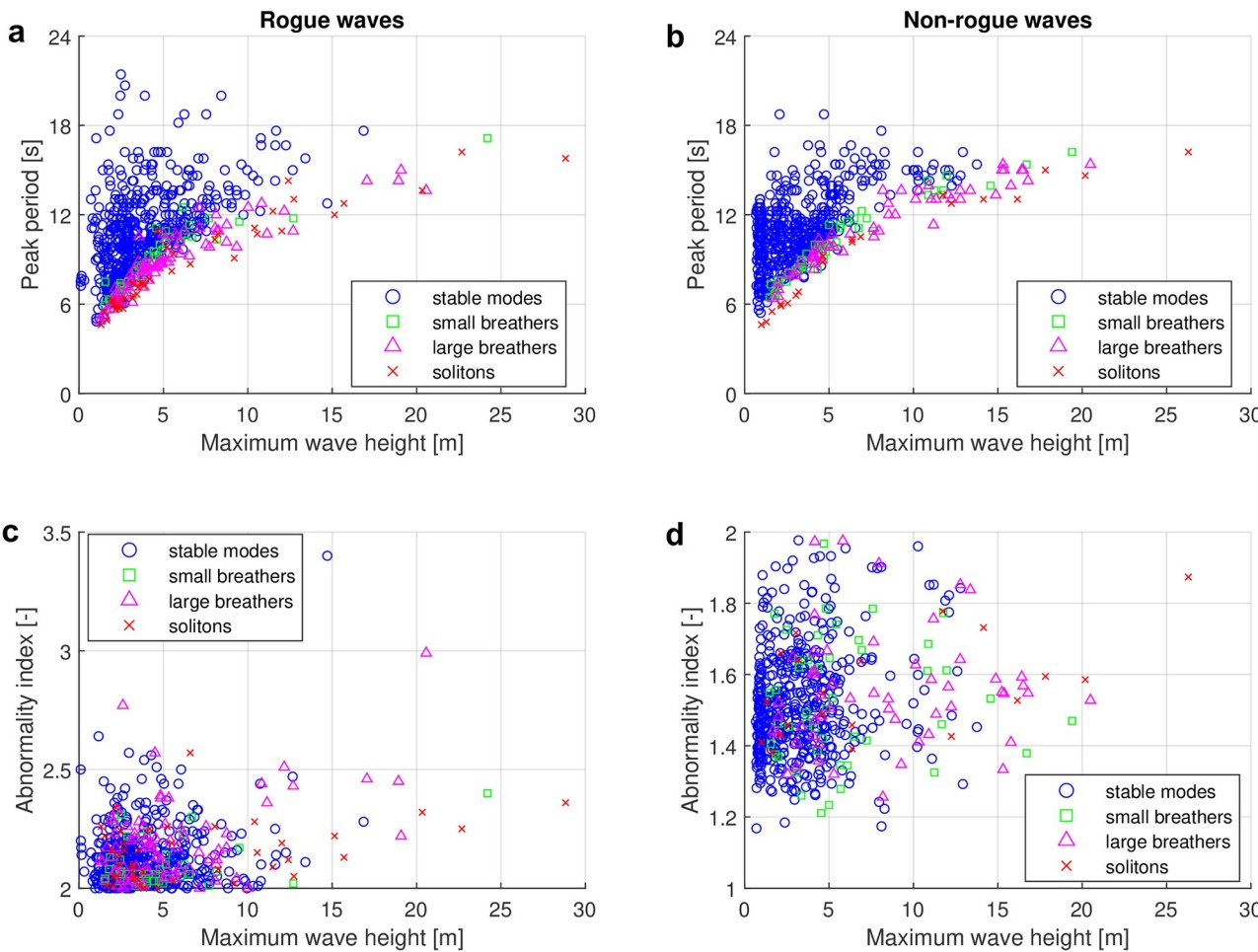

**Fig 7. Classification results of rogue and non-rogue waves based on the four types of nonlinear spectra.** (a) Scatter plot of rogue-wave samples of peak period $T_p$ over maximum wave height $H_{max}$ based on the four types of spectra. (b) Scatter plot of non-rogue wave samples of peak period $T_p$ over maximum wave height $H_{max}$ based on the four types of spectra. (c) Scatter plot of rogue-wave samples of abnormality index $AI$ over maximum wave height $H_{max}$ based on the four types of spectra. (d) Scatter plot of non-rogue wave samples of abnormality index $AI$ over maximum wave height $H_{max}$ based on the four types of spectra. The four types of spectra are stable-mode type, small-breather type, large-breather type and soliton type shown as blue circles, green squares, purple triangles and red crosses, respectively.

to evolve in the ocean. Otherwise, the waves will break when the ratio of wave height to wavelength exceeds $H/L \geq 1/7$ [109]. However, it is noteworthy that for any given maximum wave height below 15 m, the nonlinear spectra with breathers or solitons show relatively low peak periods compared with those spectra with stable modes.

To investigate how specific these results are for rogue waves, a second data set of 600 selected non-rogue samples is analyzed as well. For the non-rogue data set, we first selected all samples from 2012 and then removed all rogue waves samples. In 2012, there were 4672 non-rogue samples. From those, we keep all non-rogue samples with a maximum wave height $H_{max} \geq 5$ m. There were 130 such samples. Additionally, 470 non-rogue samples were selected randomly among the remaining samples with a maximum wave height smaller than 5 m. In total, there are thus 600 non-rogue wave samples to be analysed with NLSE-NFT. The nonlinear spectra of the non-rogue samples are also classified using the four types of spectra (stable modes, small breathers, larger breathers and solitons). The results of the classification are shown in Fig 7b. The distribution of the four types has similar features for the non-rogue

samples than for the rogue wave samples. The four types of spectra from the stable mode type to the soliton type account for 79.7%, 7.8%, 9.2% and 3.3% of the samples, respectively. The sample proportion is likely not completely representative because it is not completely randomly selected. But it is noteworthy that all non-rogue spectra with maximum wave heights larger than 14 m are type 2, type 3 or type 4 spectra with unstable modes. This suggests that the existence of unstable modes in nonlinear spectra is related to large waves in real ocean no matter if they are rogue or non-rogue waves.

In Fig 7c, we show a scatter plot of the abnormality index (*AI*) over the maximum wave height $H_{max}$ with respect to the four types of nonlinear spectra for the rogue-wave data. We expected that the nonlinear spectra with breathers or solitons may cluster at higher values of the *AI*. However, we found that there are no clear differences for the four types of spectra in this representation. Fig 7d shows a similar scatter plot for the non-rogue wave data. Also here there is no clear evidence for the four types of spectra. The non-rogue samples with maximum wave height larger then 15 m ranging between an *AI* of 1.32 and 1.88 are identified as breathers and solitons. This suggests that the existence of unstable modes in nonlinear spectra contributes to the formation of large waves in general, instead of only rogue waves.

The relationship between $H_{max}$ and $T_p$ in Fig 7a and 7b appears to be similar to a square root for large breathers and solitons. Motivated by that wave steepness is proportional to the ratio of wave height and squared period, we plot the square of the peak period over the maximum wave height in Fig 8. There is no apparent correlation for stable mode types. For spectra of unstable types, the squared peak period however does increase approximately linearly with the maximum wave height, where the slope of the linear fit depends on the exact type (small breather, large breather or soliton). Similar findings are made in the non-rogue wave case. However, the slopes of the linear fits are slightly different. Since most stable type samples are located far above the unstable type samples, the presence of a unstable dominant mode in the nonlinear spectrum is found to indicate increased wave steepness.

It was already mentioned in the introduction that the nonlinear spectrum will likely not stay constant during propagation at the site due to unaccounted factors such as directional spreading, which is present in our data (see Fig 3d and 3h). To see if directional spreading influences the classification, we have repeated the classification using only the first quartile

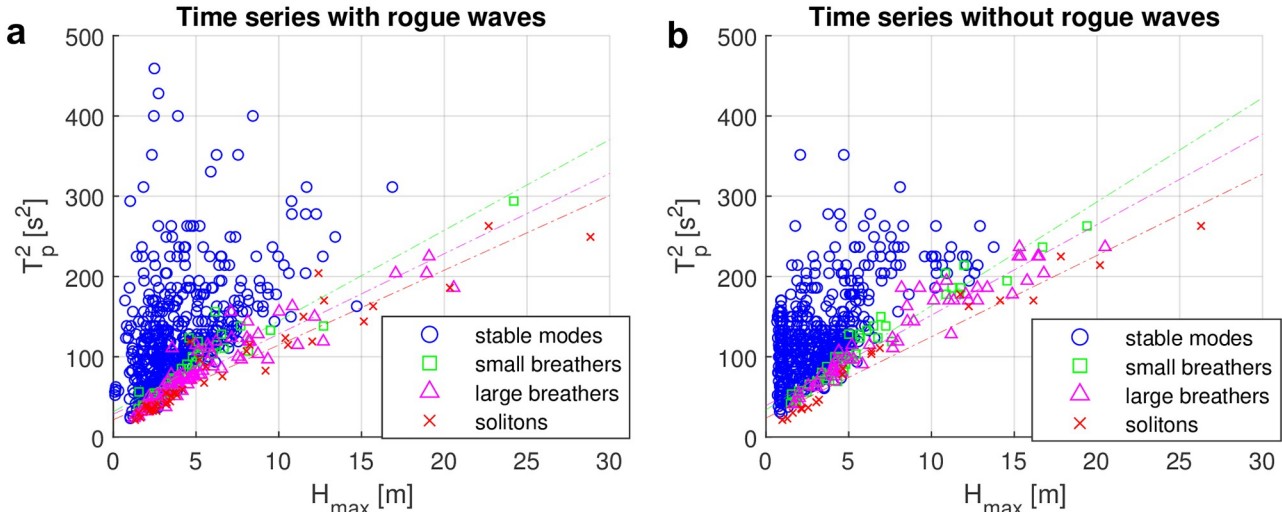

**Fig 8. Squared peak period vs maximum wave height for nonlinear spectra of the four types.** (a) Time series with rogue waves (b) Time series without rogue waves

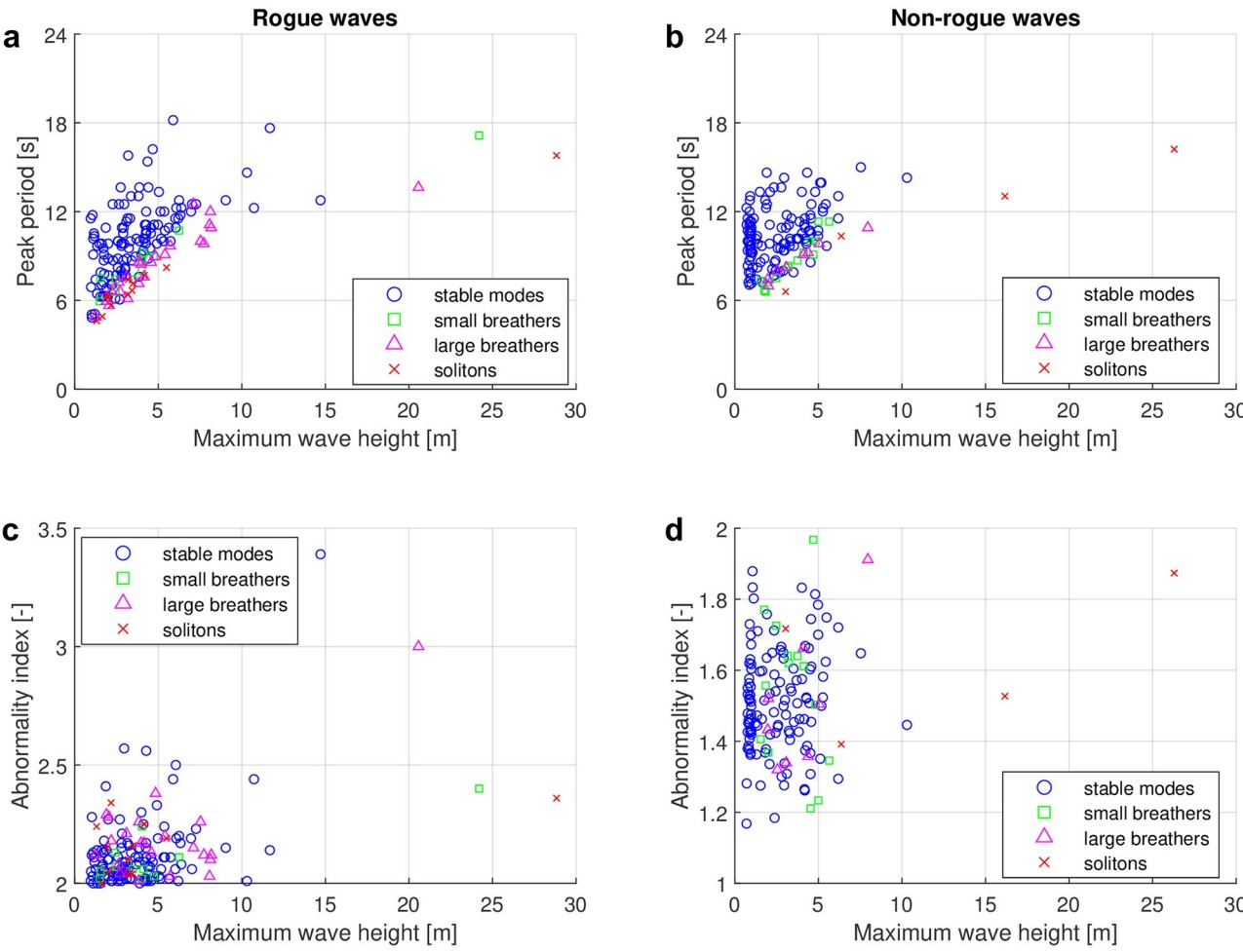

**Fig 9. Classification results of rogue and non-rogue waves with low directional spreading.** This figure presents the same information as Fig 7, but only the first quartile of the rogue waves with the lowest directional spreading have been kept for each class.

with the lowest directional spreading for each class. The results are shown in Fig 9. When comparing Figs 7 and 9, we find that the overall distributions appear to be quite similar.

We furthermore investigated the correlation of the one- and two-dimensional Benjamin-Feir indices with the maximum wave height for the four types of nonlinear spectra. The results are shown in Fig 10. In Fig 10a, we plot $BFI_{1D}$ over the maximum wave height for the rogue wave samples of each type. All stable mode spectra have $BFI_{1D}$ values below 0.3. The small breather and large breather rogue wave samples exhibit higher $BFI_{1D}$ values in the range from 0.1 to 0.4. The soliton spectra can reach the highest $BFI_{1D}$, with values up to 0.55. The four types of nonlinear spectra are somewhat separated on the scatter plot. Similar findings are however also observed for the non-rogue wave samples in Fig 10b. Since the $BFI_{1D}$ values of non-rogue wave samples appear to be even slightly larger, high values of $BFI_{1D}$ do not indicate rogue waves at the measurement site. It is also consistent with the findings in [16], where for another data set of real-world it was found that $BFI_{1D}$ is only a weak predictor for rogue waves. The mean $BFI_{1D}$ values presented in Table 1 confirm the visual impression that $BFI_{1D}$ is slightly larger for non-rogue samples with unstable types (small or larger breather, soliton). For stable mode spectra, $BFI_{1D}$ is almost identical for rogue and non-rogue samples. We furthermore observe that the mean $BFI_{1D}$ is increasing with the type of nonlinear spectrum for

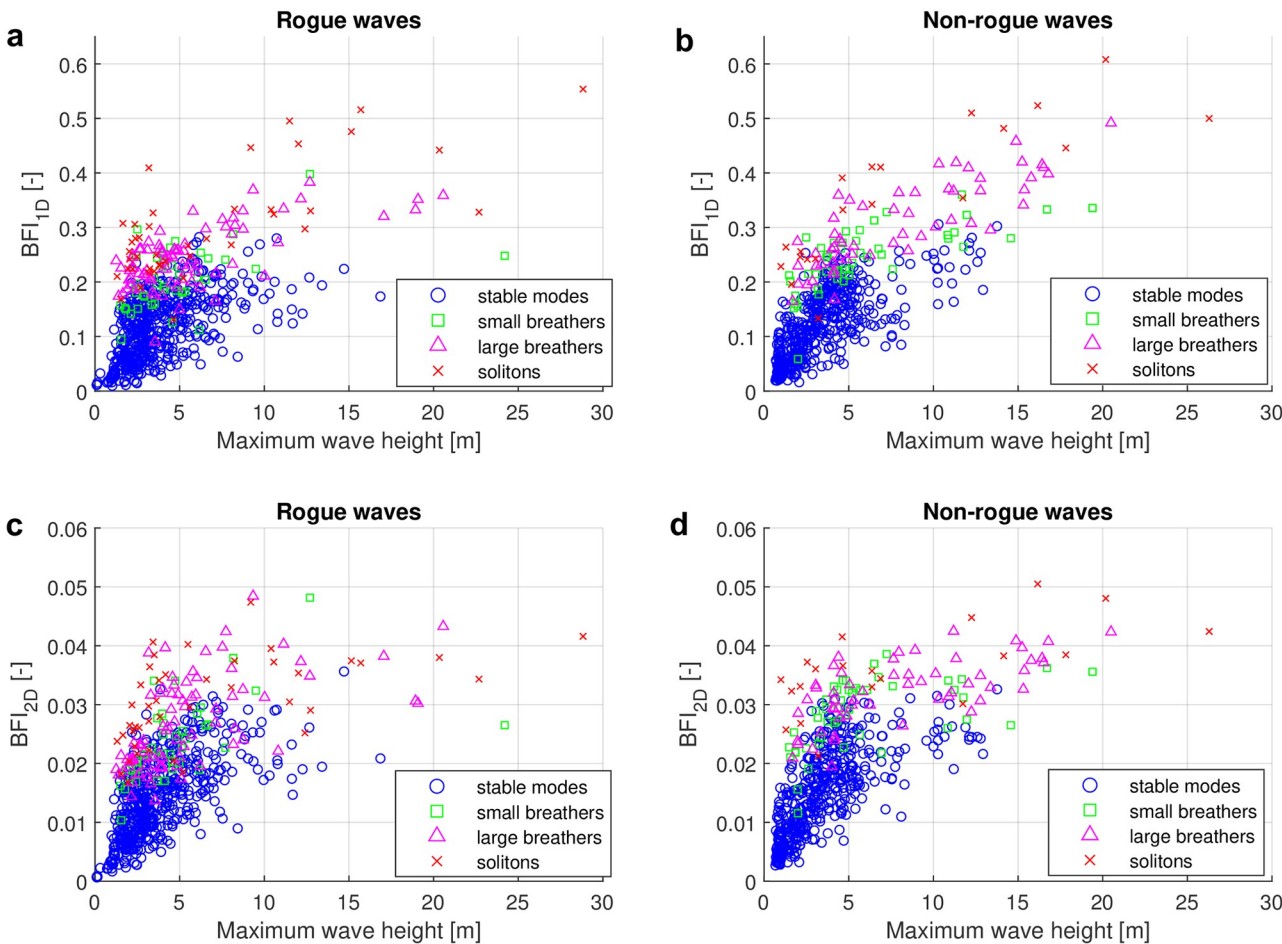

**Fig 10. Four types of nonlinear spectra and their Benjamin-Feir index.** (a) Scatter plot of rogue-wave samples of one-dimensional Benjamin-Feir index $BFI_{1D}$ over maximum wave height $H_{max}$ based on the four types of spectra. (b) Scatter plot of non-rogue wave samples of two-dimensional Benjamin-Feir index $BFI_{2D}$ over maximum wave height $H_{max}$ based on the four types of spectra. (c) Scatter plot of rogue-wave samples of one-dimensional Benjamin-Feir index $BFI_{1D}$ over maximum wave height $H_{max}$ based on the four types of spectra. (d) Scatter plot of non-rogue wave samples of two-dimensional Benjamin-Feir index $BFI_{2D}$ over maximum wave height $H_{max}$ based on the four types of spectra.

both rogue and non-rogue waves. That is, the mean $BFI_{1D}$ value is increasing as we proceed from stable modes to small breathers, larger breathers and finally solitons.

We repeated the analysis with the two-dimensional $BFI_{2D}$. The results are shown in Fig 10c and 10d. While the absolute $BFI_{2D}$ values are lower than the corresponding $BFI_{1D}$ values because of the denominator in Eq (17), we still observe that the $BFI_{2D}$ values of stable modes are on average smaller than those of unstable modes (small and large breathers, solitons) in the

**Table 1. Comparison of the mean one- and two-dimensional BFI between time series of rogue waves and non-rogue waves for the four types of nonlinear spectra.**

| Time series of rogue waves | | | Time series of non-rogue waves | | |
|---|---|---|---|---|---|
| **Types** | **Mean of $BFI_{1D}$** | **Mean of $BFI_{2D}$** | **Types** | **Mean of $BFI_{1D}$** | **Mean of $BFI_{2D}$** |
| **Stable mode** | 0.1175 | 0.0143 | **Stable mode** | 0.1161 | 0.0145 |
| **Small breather** | 0.2053 | 0.0235 | **Small breather** | 0.2431 | 0.0285 |
| **Large breather** | 0.2448 | 0.0266 | **Large breather** | 0.3077 | 0.0323 |
| **Soliton** | 0.2997 | 0.0306 | **Soliton** | 0.3559 | 0.0360 |

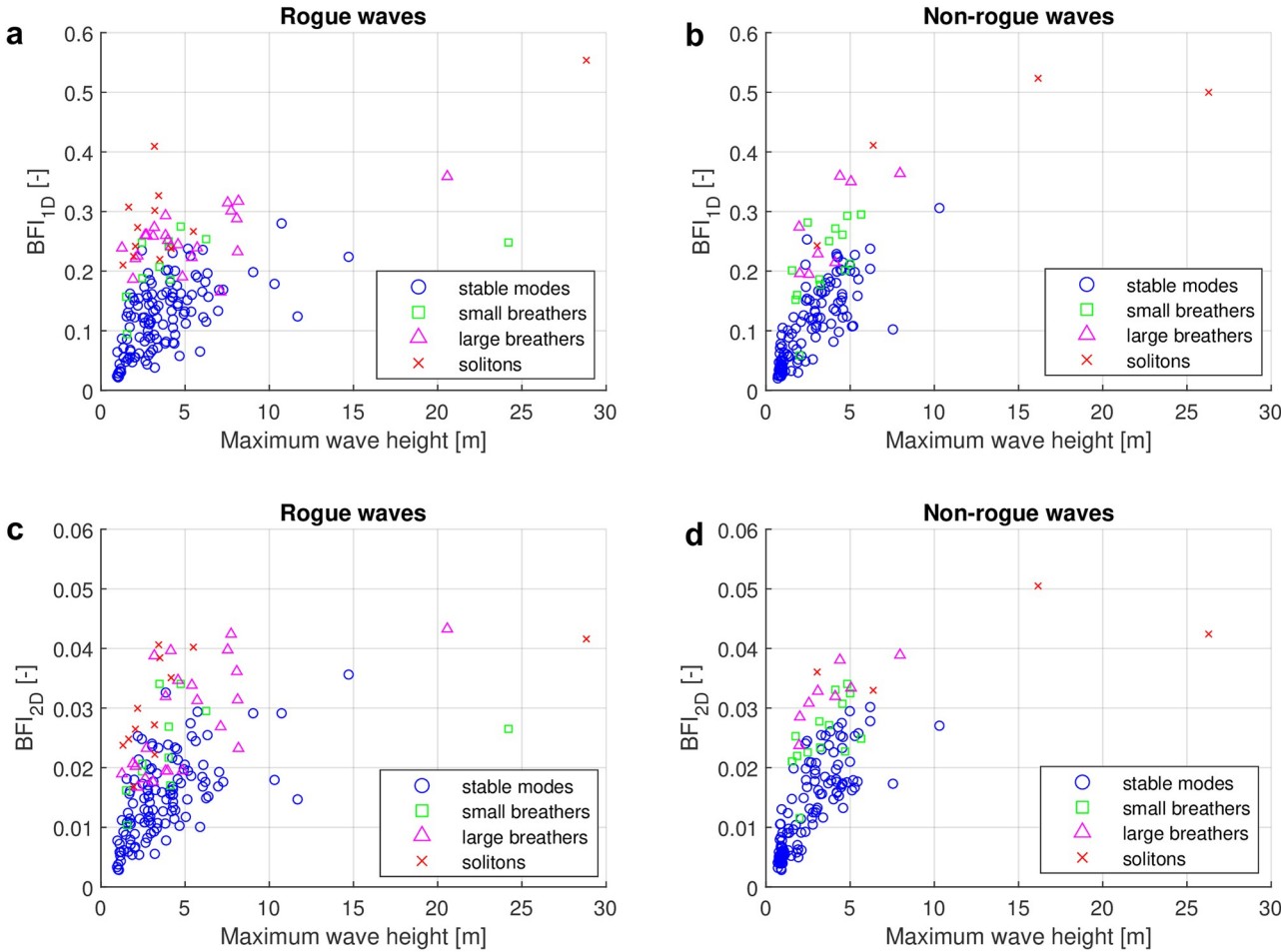

**Fig 11. Four types of nonlinear spectra and their Benjamin-Feir index with low directional spread.** This figure presents the same information as Fig 10, but only the first quartile of the rogue waves with the lowest directional spreading have been kept for each class.

rogue wave case. The stable mode spectra exhibit lower $BFI_{2D}$'s in the range of 0 to 0.036, while the $BFI_{2D}$ values for small breathers, large breathers, and soliton spectra fall within the range of 0.01 to 0.05. Within the group of unstable modes, small breathers, large breathers, and soliton spectra can no longer be clearly distinguished. The mean values of $BFI_{2D}$ in Table 1 however reveal that the mean values of $BFI_{2D}$ are still increasing as we progress from stable modes to small breathers, large breathers and solitons. Similar findings are observed for the non-rogue wave samples, as depicted in Fig 10d. Large $BFI_{2D}$ values therefore again do not indicate the existence of rogue waves.

To see if directional spreading influences the relation between the BFI's for the four types of nonlinear spectra, we finally repeated the analysis using only the first quartile with the lowest directional spreading for each class, as shown in Fig 11. When comparing Figs 10 and 11, we observe that the two figures exhibit a considerable degree of similarity. The directional spreading does not alter the distribution of the four types of spectra significantly.

## Relationship of unstable modes in nonlinear spectra to rogue waves

In this section, we investigate whether the unstable mode (i.e., small breather, large breather, or soliton) with the largest amplitude in the nonlinear spectrum can be attributed to rogue

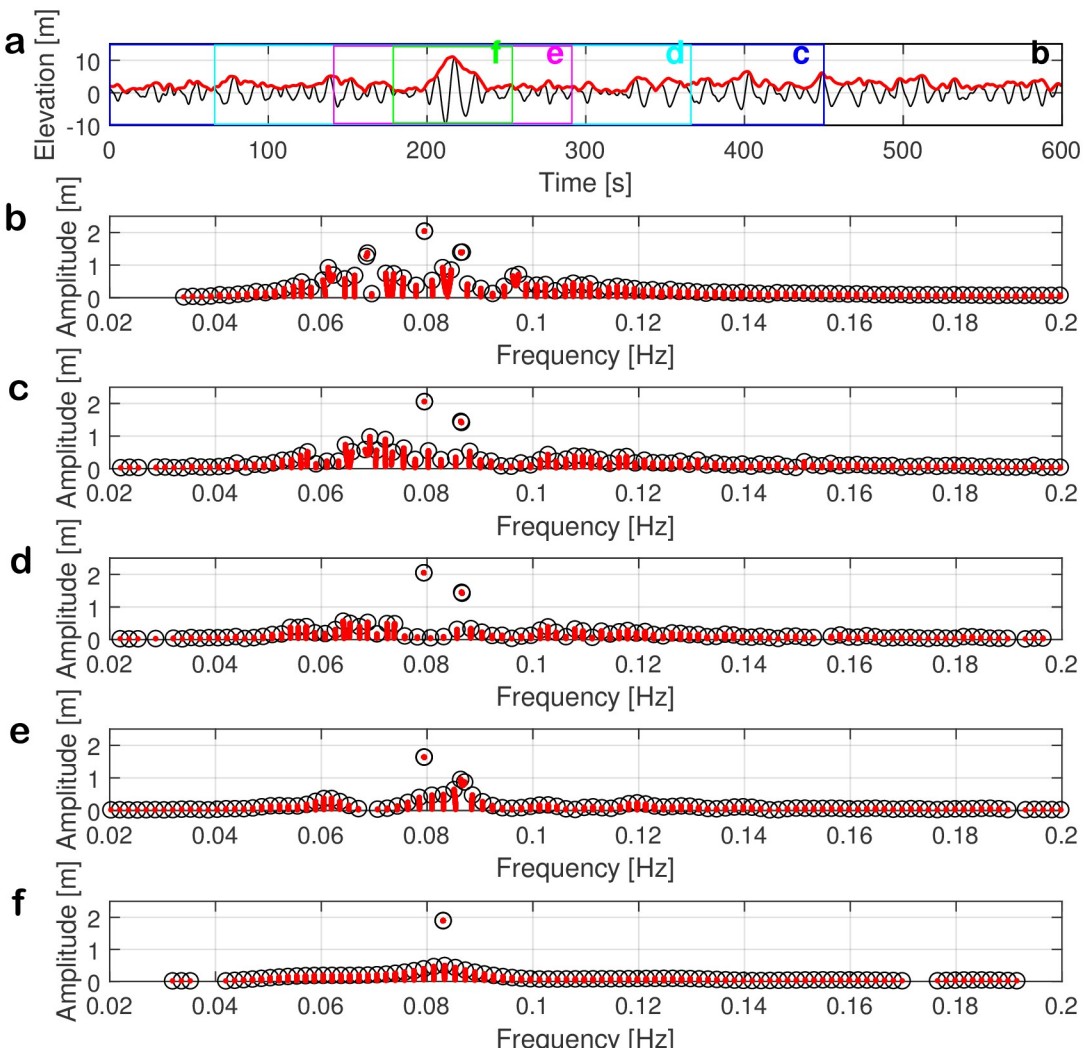

**Fig 12. Relationship of unstable modes to rogue waves.** (a) The rogue-wave data is measured from 06:00h to 06:10h on 29 July 2017 with the maximum wave height $H_{\max} = 20.34$ m, the significant wave height $H_s = 8.76$ m, the peak wave period $T_p = 13.64$ s, and the abnormality index $AI = 2.32$. The surface-wave elevation is shown as the black line and the magnitude of the complex envelope calculated by Hilbert transform is shown as red line. The original signal is cut into different time lengths of 600 s, 450 s, 300 s, 150 s, and 75 s, which are labelled from b to f. (b-f) Nonlinear spectra corresponding to the time series from b to f.

waves in time series. We select the time series with a rogue wave shown in Fig 12a as an example. The time series was measured from 06:00h to 06:10h on 29 July 2017 during an extreme sea state. The maximum wave height is $H_{\max} = 20.34$ m, the significant wave height is $H_{1/3} = 8.76$ m, the peak wave period is $T_p = 13.64$ s, and the abnormality index is $AI = 2.32$. In order to determine which nonlinear modes contribute to the rogue wave, we propose a new windowing approach. We cut the localized structures of rogue wave with 450 s, 300 s, 150 s and 75 s (see Fig 12a). Outside these time frames, the elevation of the time series is set to zero. The entire duration of the sample keeps 600 s. The window is initially positioned at the center of the rogue wave. If the window extends beyond the bounds of the time series, it is subsequently adjusted by moving it left and right within the time series. Similar to zero-padding for the conventional FFT, this approach ensures that the nonlinear frequency resolution stays the same when the NLSE-NFT is computed. We remark that the popular approach of cutting out short

parts of the time series and analyzing them with a shorter signal period [83] did not allow us to localize nonlinear modes. Note that in our approach, the period of the signal is not reduced.

Fig 12a shows the localized structures b to f outside which the time series in Fig 12a was set to zero before computing the nonlinear spectra, which are presented in Fig 12b–12f. These plots only show the center part of the spectra with frequencies from 0.02 Hz to 0.2 Hz, including the dominant frequencies. The nonlinear spectrum of the original time series is shown in Fig 12b. There are five unstable modes: one soliton with coordinates (0.079 Hz, 2.05 m) and four breathers. Two unstable modes have larger amplitudes with coordinates (0.068 Hz, 1.27m), (0.069 Hz, 1.37 m) and (0.086 Hz, 1.4 m), (0.087 Hz, 1.4 m) that satisfied the criterion of being larger than 120% than the maximum amplitude of the stable modes. The other two smaller breathers are at the coordinates (0.061 Hz, 0.92 m), (0.062 Hz, 0.7 m) and (0.096 Hz, 0.66 m), (0.097 Hz, 0.72m).

The nonlinear spectra of the partially zeroed time-series in the Figs. Fig 12c–12f show that most of the spectral amplitudes reduce. The number of unstable modes in the nonlinear spectrum in Fig 12c is already reduced. While the soliton and the large breather with the higher frequencies still exist, the large breather with the smaller frequency changed into a small breather. Two smaller breathers from the original spectrum of Fig 12b turn into small stable modes. A remarkable finding here is that the soliton in Fig 12c keeps nearly the same frequency and amplitude as in Fig 12b, indicating that the cut-off fragment from 450 s to 600 s in the original time series does not contain wave components that correspond to this soliton. In other words, this soliton exists in both time series.

Similarly, this soliton is still retained in the spectra of Fig 12d and 12e, whereas the other spectral components change owing to the reduced window length of 300 s and 150 s, respectively, outside which the time series is set to zero. When the window length is 75 s, it only contains a part of the rogue wave. The corresponding nonlinear spectrum in Fig 12f is dominated by the soliton with nearly the same amplitude as in the previous spectra, but with a slightly shifted frequency. This shows that the largest soliton amplitude in the nonlinear spectrum is related to the rogue wave at 218 s in the time series of Fig 12a.

The other three types was tested and discussed further in the S1 File. There we found that the maximum unstable modes in a type 3 spectrum (large-breather) can be attributed to the rogue wave as well, even though this has only been possible up to the second strongest truncation. In contrast, we were not able to attribute type 1 spectrum (stable-mode). This finding is expected for stable modes since sine waves and Stokes waves are not localized. We were also not able to attribute type 2 spectrum (small-breather) unstable modes to rogue waves for the smaller window sizes.

In the S1 File, these findings have been tested further for additional 20 rogue waves samples (five for each type). The finding that for soliton type spectra, the soliton is found in the rogue wave has been confirmed in all but in one case. Only in the exceptional case, the soliton disappeared at maximum truncation. The attribution of large breather type spectra in contrast turns out to be more difficult as there are often several large breathers in the nonlinear spectrum. The largest one cannot always be attributed to the rogue wave.

Now that we know that the dominant nonlinear modes of the breather and soliton types are located at the rogue waves, the question of how much they contribute to the rogue waves arises. We therefore relate the amplitude of the largest nonlinear mode to the maximum wave height and maximum crest height of the time series with rogue waves in a scatter plot in Fig 13. The comparison does not contain stable modes, as those were not found to be localized. We observe that for both small and large breather modes, the relation between their amplitudes and the rogue wave height and crest height is approximately linear. The amplitudes of the dominant mode are in the range of roughly 2.5% to 10% of the maximum wave height, and

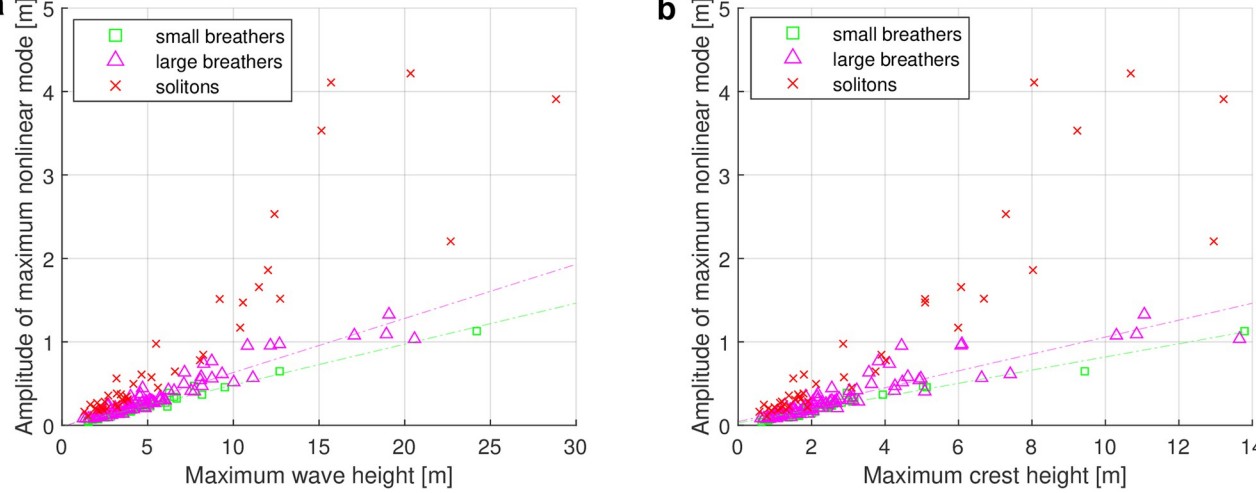

**Fig 13. Relationship of the largest nonlinear mode to wave parameters.** (a) Scatter plot of the largest nonlinear mode over maximum wave height. (b) Scatter plot of the largest nonlinear mode over maximum crest height.

roughly 5% to 20% of the maximum crest height in these cases. The picture is different when the dominant mode is of the soliton type, with the amplitudes of maximum soliton ranging from approximately 7% to 27% of the maximum wave height, and approximately 14% to 51% of the maximum crest height in these cases. Notably, for maximum wave heights larger than 15 m, the amplitude of dominant modes of the soliton type can arrive up to one quarter of the maximum wave height. For crest heights larger than 8 m, the amplitude dominant modes of the soliton type can account for half of the maximum crest height. The two figures illustrate similar results. It is worth noting that the amplitudes of the maximum unstable modes are represented by the envelope, which is entirely above the sea surface. This characteristic facilitates a more straightforward comparison with crest height.

## Summary and discussion

In the results section, we have classified the nonlinear spectral portraits of 663 rogue wave samples that were measured by the Taitung Open Ocean buoy in the Philippine Sea from 2006 to 2017, using the periodic NLSE-NFT. For comparison, we also analyzed 600 selected non-rogue wave samples from 2012. The four different types of spectra were stable modes (i.e., sine or Stokes waves), small breathers, large breathers and solitons. We found that the majority of the rogue wave samples (75.1%) were stable mode spectra.

This finding is in line with with other studies such as [16, 108], where the majority of rogue waves were found to be generated from linear mechanisms. Furthermore, it complements the different observations in [54, 73], where dominant modes of the stable and breather type, respectively, were attributed to rogue waves.

Another interesting finding is that a large portion (90.9%) of the rogue wave samples with maximum wave heights larger than 15 m are of the breather or soliton type. When analyzing the non-rogue samples a similar picture was found. All non-rogue wave samples with maximum wave heights larger than 14 m were again of breather and soliton types. This suggests that dominant breather and soliton components in the nonlinear spectrum are in general representative for (the complex envelopes of) time series with large maximum wave heights. Furthermore, we found that they can also be associated with increased wave steepness.

By studying the relations between the one- and two-dimensional BFIs and the maximum wave height, we found that unstable modes have higher BFI's than stable modes in all cases (1D and 2D, rogue and non-rogue). This seems plausible because both BFIs can be linked to kurtosis under certain assumptions (i.a. narrowbandedness), which in turn is linked to nonlinear interactions [3, 103]. Therefore, our results seem to confirm that sea states with an unstable dominant mode are more nonlinear than sea states with a stable dominant mode. Moreover, we found that the distributions of the BFIs are similar for rogue and non-rogue waves. If the BFIs were correlated with the rogue wave probability, one would expect higher BFIs for the rogue wave data. Since that does not appear to be the case here, our results do *not* suggest a connection between the BFIs and the rogue wave probability. This is contrast to the unidirectional case, where $BFI_{1D}$ is known to correlate with the rogue wave probability [3, 5, 33]. The finding is however in line with other field measurements [16].

The one-dimensional BFI can also be related to the modulational instability, both from a stochastic and a deterministic point of view. For random waves, Alber [110] has shown that narrowband Gaussian random deep-water wavetrains can be modulationally unstable if $BFI_{1D}$ > 1. Numerical simulations indicate that modulational instability can also occur for $BFI_{1D}$ < 1, where the effect is gradually decreasing [3, p. 866]. The largest $BFI_{1D}$'s in our data set are smaller than 0.6, while the large majority of rogue wave events occurred in the regime $BFI_{1D}$ < 0.4. It is also known that the modulational instability is weakened in directional wave fields [103]. Since the directional spreading is strong in our data (see Fig 3d and 3h), it seems unlikely that the modulational instability played a significant role for the formation of rogue waves. From a deterministic point of view, we know that modulational instability is only possible if the nonlinear spectrum has a degenerate double point in the main spectrum [56]. Due to the large water depth, all our rogue waves occurred in the focusing regime of the NLSE, in which the modulational instability can occur. However, since any small random perturbation will split up the double points, the probability of actually measuring a modulationally unstable wave train (in the deterministic sense) is zero. Consequently, no degenerate points were observed in our data. As an alternative, the so-called splitting distance has been proposed in order to instead estimate how close a generic nonlinear spectrum is to one with a degenerate point [44, 45]. To the best of our knowledge, the splitting distance has not yet been evaluated on real-world measured rogue waves. However, in simulations of the NLSE it was found that it does not correlate with maximum wave height, while the amplitude of the dominant nonlinear mode considered in this work does [46].

Using a new procedure to localize nonlinear modes, the largest nonlinear mode of exemplary rogue wave samples of the soliton type were shown to be localized in the rogue wave. This is an interesting difference to conventional Fourier analysis, where individual modes (i.e. sinusoidal waves) in themselves are never localized and thus always have an impact on the complete time series. We were not able to localize the largest modes of exemplary rogue wave samples of the small breather and stable mode types in the corresponding rogue waves. The results for large breather modes were mixed. We suspect that smaller breathers are less localized. Stable modes (i.e. sine or Stokes waves) are furthermore not localized at all. Hence, this finding seems plausible.

By relating the amplitude of the dominant modes to the height of the corresponding rogue waves, we found that the dominant nonlinear modes of the localized breather and soliton types in general only contribute a small fraction of the rogue waves total wave height. This overall picture is consistent with other studies such as [54, 57], where the dominant nonlinear mode had to interact with other wave components to form the rogue wave. It often seems to push already large waves over the rogue wave threshold. Our data however includes a noteworthy exception to this general pattern, for rogue waves of the soliton type with large wave

heights. In those cases, the dominant soliton component could contribute more than half of the total height of the rogue wave in extreme cases. Here, it is important to keep in mind that the soliton components are envelope solitons (because the NLSE is a envelope equation, see the Section Data normalization of deep water waves for the nonlinear Fourier transform). Through the lens of surface elevations, they actually correspond to travelling wave groups. The strong contribution of solitons (in the complex envelope) to rogue waves has also been observed in the literature [78, Figs. 1C and 2C], where an (vanishing) NLSE-NFT-based analysis showed the same pattern for two measured rogue waves.

Finally, we also observed that rogue wave samples with unstable modes have low peak periods when compared to stable mode samples with the same maximum wave heights. Furthermore, there is no clear relation between the abnormality index (AI) and four types of nonlinear spectra. The relation between the squared peak period and the maximum wave height has however been found to be approximately linear for nonlinear spectra of the breather and soliton types.

In conclusion, this study applied the NLSE-NFT to classify rogue waves that were measured in the Philipine sea under deep water conditions. It is the first time that the NLSE-NFT has been systematically applied to study a large number of real-world rogue waves in deep water. Prior NLSE-NFT-based studies of individual rogue wave events had resulted in very different nonlinear spectra, ranging from dominating stable modes over breather to soliton modes. Our study shows that all three cases can occur at a single measurement site. We found stable modes are most typical for rogue waves in general, but that rogue waves with large wave heights are mostly of the large breather or soliton type. We furthermore found that while the contribution of the dominant nonlinear mode to the rogue wave is in general small, it can contribute up to 50% for large rogue waves of the soliton type.

While the classification results for non-rogue waves show no significant differences to the rogue wave samples, we point out that the difference between nonlinear rogue and non-rogue spectra might be qualitative instead of quantitative. We only classified the type of spectrum, but the difference could also be in the arrangement of the components. The sizes of certain gaps have been found to relate to the probability of seeing a rogue wave under certain circumstances in simulations of the NLSE in Ref. [44, 45]. A similar observation has been made recently in the analysis of rogue waves that were measured under shallow water conditions in the north sea [57].

We finally remark again that in particular due to the presence of directional spreading the nonlinear spectra most likely would change under propagation at the measurement site. While our classification results did not change noticeably when only the lowest quartile with the least directional spreading was considered for each type of rogue waves, the impact of directionality on nonlinear spectra remains not well understood.

## Supporting information

**S1 File. Localization of nonlinear modes for periodic nonlinear Fourier analysis of rogue waves.** This file contains additional information for identifying rogue waves from the solutions of the NLSE-NFT.
(PDF)

## Acknowledgments

We want to thank the editor, referees, and everyone in our discussion group for their helpful comments. Your input has made this journal publication much better.

## Author Contributions

**Conceptualization:** Yu-Chen Lee.

**Data curation:** Yu-Chen Lee, Dong-Jiing Doong.

**Formal analysis:** Yu-Chen Lee, Sander Wahls.

**Funding acquisition:** Yu-Chen Lee, Sander Wahls.

**Investigation:** Yu-Chen Lee, Sander Wahls.

**Methodology:** Yu-Chen Lee, Sander Wahls.

**Project administration:** Dong-Jiing Doong, Sander Wahls.

**Software:** Yu-Chen Lee, Sander Wahls.

**Supervision:** Markus Brühl, Sander Wahls.

**Validation:** Yu-Chen Lee, Markus Brühl, Sander Wahls.

**Visualization:** Yu-Chen Lee, Sander Wahls.

**Writing – original draft:** Yu-Chen Lee.

**Writing – review & editing:** Yu-Chen Lee, Markus Brühl, Sander Wahls.

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
