## [Decision Letter · Decision Letter 0]

16 Oct 2023

PONE-D-23-29589Nonlinear Fourier Classification of 663 Rogue Waves Measured in the Philippine SeaPLOS ONE

Dear Dr. Lee,

Thank you for submitting your manuscript to PLOS ONE. A review of the paper, produced by one of top experts in the area, suggests that an essential revision of the paper is necessary.

We look forward to receiving your revised manuscript.

Kind regards,

Boris Malomed

Academic Editor

PLOS ONE

Journal Requirements:

3.Thank you for stating the following financial disclosure: 

"This work was supported by the European Research Council (ERC) under the European Union’s Horizon 2020 Research and Innovation Programme (grant agreement No 716669), and also from 2019 Key Fields Scholarship Program, the Ministry of Education, Republic of China (R.O.C.), Taiwan."  

"This work was supported by the European Research Council (ERC) under the European 572

Union’s Horizon 2020 Research and Innovation Programme (grant agreement No 573

716669), and also from 2019 Key Fields Scholarship Program, the Ministry of 574

Education, Republic of China (R.O.C.), Taiwan."

"This work was supported by the European Research Council (ERC) under the European Union’s Horizon 2020 Research and Innovation Programme (grant agreement No 716669), and also from 2019 Key Fields Scholarship Program, the Ministry of Education, Republic of China (R.O.C.), Taiwan."

Reviewers' comments:

Reviewer's Responses to Questions

**Comments to the Author**

1. Is the manuscript technically sound, and do the data support the conclusions?

Reviewer #1: Yes

2. Has the statistical analysis been performed appropriately and rigorously? 

Reviewer #1: Yes

3. Have the authors made all data underlying the findings in their manuscript fully available?

Reviewer #1: No

4. Is the manuscript presented in an intelligible fashion and written in standard English?

Reviewer #1: Yes

5. Review Comments to the Author

Reviewer #1: Please see the attached report

.............................................................................................................................................................................................

6. PLOS authors have the option to publish the peer review history of their article (what does this mean?). If published, this will include your full peer review and any attached files.

Reviewer #1: No

---

## [Author Response · Author response to Decision Letter 0]

6 Dec 2023

Dear Professor Malomed,

I wanted to express our gratitude for your dedicated efforts in overseeing the publication process of our manuscript in PLOS ONE. Your guidance and support have been invaluable, and we appreciate your commitment to advancing scholarly work in this field.

In response to the constructive feedback provided during the review process, we have diligently revised the manuscript. Enclosed, please find the updated version, along with a marked-up copy illustrating the changes made, and a detailed response to the reviewers' comments.

Your time and expertise are highly valued, and we genuinely appreciate your ongoing support throughout this publication journey. Please let us know if there are any additional steps required or if further information is needed to facilitate the review of the revised manuscript.

Thank you once again for your dedication to the advancement of research and your contributions to the scholarly community.

Best regards,

Yu-Chen

---

## [Decision Letter · Decision Letter 1]

2 Jan 2024

PONE-D-23-29589R1Nonlinear Fourier Classification of 663 Rogue Waves Measured in the Philippine SeaPLOS ONE

Dear Dr. Lee,

The review of your revised paper suggests that a minor additional  revision is needed. Therefore, I invite you to submit a revised version of the manuscript that addresses the points raised in the latest review.

We look forward to receiving your revised manuscript.

Kind regards,

Boris Malomed

Academic Editor

PLOS ONE

Journal Requirements:

Reviewers' comments:

Reviewer's Responses to Questions

**Comments to the Author**

1. If the authors have adequately addressed your comments raised in a previous round of review and you feel that this manuscript is now acceptable for publication, you may indicate that here to bypass the “Comments to the Author” section, enter your conflict of interest statement in the “Confidential to Editor” section, and submit your "Accept" recommendation.

Reviewer #1: All comments have been addressed

2. Is the manuscript technically sound, and do the data support the conclusions?

Reviewer #1: Yes

3. Has the statistical analysis been performed appropriately and rigorously? 

Reviewer #1: Yes

4. Have the authors made all data underlying the findings in their manuscript fully available?

Reviewer #1: No

5. Is the manuscript presented in an intelligible fashion and written in standard English?

Reviewer #1: Yes

6. Review Comments to the Author

Reviewer #1: Please see the attached file.

7. PLOS authors have the option to publish the peer review history of their article (what does this mean?). If published, this will include your full peer review and any attached files.

Reviewer #1: No

---

## [Author Response · Author response to Decision Letter 1]

14 Mar 2024

Dear Professor Malomed,

I wanted to express our gratitude for your dedicated efforts in overseeing the publication process of our manuscript in PLOS ONE. Your guidance and support have been invaluable, and we appreciate your commitment to advancing scholarly work in this field. 

In response to the constructive feedback provided during the review process, we have diligently revised the manuscript. Enclosed, please find the updated version with version 2, along with a marked-up copy illustrating the changes made, and a detailed second response to the reviewers' comments.

Your time and expertise are highly valued, and we genuinely appreciate your ongoing support throughout this publication journey. Please let us know if there are any additional steps required or if further information is needed to facilitate the review of the revised manuscript.

Thank you once again for your dedication to the advancement of research and your contributions to the scholarly community.

Best regards,

Yu-Chen

---

## [Editor Report · Decision Letter 2]

20 Mar 2024

Nonlinear Fourier Classification of 663 Rogue Waves Measured in the Philippine Sea

PONE-D-23-29589R2

Dear Dr. Lee,

We’re pleased to inform you that your manuscript has been judged scientifically suitable for publication and will be formally accepted for publication once it meets all outstanding technical requirements.

Kind regards,

Boris Malomed

Academic Editor

PLOS ONE
---

## [Editor Report · Acceptance letter]

26 Mar 2024

PONE-D-23-29589R2 

PLOS ONE

Dear Dr. Lee, 

I'm pleased to inform you that your manuscript has been deemed suitable for publication in PLOS ONE. Congratulations! Your manuscript is now being handed over to our production team.

Kind regards, 

on behalf of

Prof. Boris Malomed 

Academic Editor

PLOS ONE